# Simultaneously discovering the fate and biochemical effects of pharmaceuticals through untargeted metabolomics

Tara J. Bowen [1], Andrew D. Southam [1,2], Andrew R. Hall[3], Ralf J. M. Weber [1,2], Gavin R. Lloyd [2], Ruth Macdonald[4], Amanda Wilson [5], Amy Pointon [3] & Mark R. Viant [1,2] ✉

Untargeted metabolomics is an established approach in toxicology for characterising endogenous metabolic responses to xenobiotic exposure. Detecting the xenobiotic and its biotransformation products as part of the metabolomics analysis provides an opportunity to simultaneously gain deep insights into its fate and metabolism, and to associate the internal relative dose directly with endogenous metabolic responses. This integration of untargeted exposure and response measurements into a single assay has yet to be fully demonstrated. Here we assemble a workflow to discover and analyse pharmaceutical-related measurements from routine untargeted UHPLC-MS metabolomics datasets, derived from in vivo (rat plasma and cardiac tissue, and human plasma) and in vitro (human cardiomyocytes) studies that were principally designed to investigate endogenous metabolic responses to drug exposure. Our findings clearly demonstrate how untargeted metabolomics can discover extensive biotransformation maps, temporally-changing relative systemic exposure, and direct associations of endogenous biochemical responses to the internal dose.

Untargeted metabolomics is increasingly applied across a wide range of fields including toxicology[1–3]. This approach uses an untargeted analytical detector, typically mass spectrometry (MS), to measure large numbers of low molecular weight biochemicals (metabolites and lipids) in a biological sample[4]. In toxicology, such information can be utilised to discover modes-of-action[3,5–7]; derive molecular points-of-departure from baseline as a means to calculate safety thresholds[3,6,8]; and group chemicals into categories based on similarities of metabolic responses to increase the reliability of read-across in chemical risk assessments[3,6,9]. A consequence of the untargeted nature of the analytical detector is that xenobiotics and related compounds are typically measured alongside endogenous metabolites[1,3]. Simultaneous measurement of the exposure xenobiotic(s) and its biotransformation products can confuse the interpretation of endogenous metabolic responses if they remain unidentified. Consequently, it is recommended to identify and discard these xenobiotic-related data prior to statistical analyses[10]. Yet measurements of xenobiotics can provide valuable information, revealing insights into the exposome, i.e., the chemical environment to which an individual has been exposed[3,11,12], and into their fate and metabolism[1,13–19]. While stable isotope and untargeted MS-based approaches have successfully discovered biotransformation products of xenobiotics[12,20–25], the integration of untargeted endogenous metabolomics data with measurements of internal dose and biotransformation to simultaneously understand the fate and effects of xenobiotics is limited[26].

[1]School of Biosciences, University of Birmingham, Edgbaston, Birmingham B15 2TT, UK. [2]Phenome Centre Birmingham, University of Birmingham, Edgbaston, Birmingham B15 2TT, UK. [3]Safety Sciences, Clinical Pharmacology and Safety Sciences, BioPharmaceuticals R&D, AstraZeneca, Cambridge, UK. [4]Animal Sciences and Technology, Clinical Pharmacology and Safety Sciences, BioPharmaceuticals R&D, AstraZeneca, Cambridge, UK. [5]Integrated Bioanalysis, Clinical Pharmacology and Safety Sciences, BioPharmaceuticals R&D, AstraZeneca, Cambridge, UK. ✉e-mail: m.viant@bham.ac.uk

Characterising the exposure to, and biotransformation of, a xenobiotic are key factors in the safety assessment of pharmaceuticals, biocides and industrial chemicals[13–19], anti-doping testing[27], and environmental risk assessment[3,11]. Internal dose measurements disclose information on a xenobiotics disposition (absorption, distribution, metabolism and elimination; ADME)[28], e.g., by confirming distribution to target organ(s), indicating the time and concentration of peak internal dose, and revealing steady-state levels (after repeated dosing)[13,14,16,18]. Such information can help to enable accurate prediction of human risk from the responses of model species[13,16–18]. Biotransformation of a xenobiotic may result in compounds that are more toxic than the parent and, consequently, the toxicity of biotransformation products to which humans are exposed—either through pharmacological intervention or unwanted exposure—must be evaluated[14,15,19]. This requires the measurement of complete biotransformation maps of xenobiotics[14,15,19]. Knowledge of biotransformation maps is also important for risk assessment of industrial chemicals, e.g., serving as a basis for chemical grouping to support read-across of toxicity data[29].

Standardised methods for measuring xenobiotic internal dose and/or biotransformation are described in guidance documents that support regulatory safety legislation for pharmaceuticals, biocides and industrial chemicals[16,18]. Methods typically rely on [$^{14}$C]-labelling (gold-standard)[30], or more recently, stable isotopic labelling ([$^{13}$C] or [$^{2}$H])[22,23], to track xenobiotic biotransformation, limiting their application to pre-defined exposures. Additionally, while these methods are not intended to provide insights into the impacts of a xenobiotic on endogenous biochemistry, the potential to acquire information simultaneously on the disposition of a xenobiotic alongside its biological impacts is tantalising. This could enable a deeper assessment of the covariance of systemic or organ-specific exposure and endogenous response, revealing otherwise undiscovered toxicological insights. Untargeted MS-based analysis of biological samples, with statistical-based data mining, offers a proven alternative strategy to biotransformation product elucidation[24] that may enable such simultaneous discovery of xenobiotic disposition and effect. Furthermore, there is an unmet need for rapidly and affordably characterising xenobiotic biotransformation in vitro, yet knowledge of such characteristics could enhance the in vitro-in vivo extrapolation of dose-response relationships, helping to promote the use of in vitro models for safety assessments and contributing to a reduced need for expensive and time-consuming in vivo toxicology studies[31,32].

The overarching aim of the current study was to demonstrate an integrative method to discover extensive xenobiotic-related data from routine untargeted metabolomics datasets and subsequently derive information on the disposition of an exposure substance, over time, obtained from the same biological samples and analytical measurements used to discover the endogenous metabolic responses. This type of analysis may be termed 'untargeted toxicokinetics'[6] (TK) or untargeted ADME/TK. The first objective was to develop a computational open-source workflow to semi-automatically discover and visualise the measurements of a xenobiotic and its biotransformation products within untargeted metabolomics datasets—here demonstrated using pharmaceuticals. Secondly, we sought to demonstrate the capability of this workflow to discover fingerprints of xenobiotic biotransformation within the plasma (with temporal resolution) and at a site of toxicity (the heart), in rats exposed to two cardiotoxins, sunitinib and KU60648. The third objective was to reveal the capacity, by association of measured internal relative doses with the endogenous biochemical responses, to uncover toxicologically relevant perturbations, including at a site of toxicity. Next, we attempted to demonstrate the capability to reveal the metabolic competencies of in vitro models by applying the workflow to sunitinib-exposed human induced pluripotent stem cell-derived cardiomyocytes (hiPSC-CMs). Our final objective was to highlight the ability to discover exposures of humans to several pharmaceuticals and, through implementation of the workflow, characterise their metabolic fate.

## Results

### Development of untargeted workflow to discover xenobiotics

Untargeted metabolomics datasets are measured to provide information on endogenous biochemistry but can also reveal insights into the fate and metabolism of xenobiotics. Here we developed a data processing and analysis workflow to discover information arising from exposure to a xenobiotic and its biotransformation products using data that can be measured routinely by state-of-the-art ultra-high performance liquid chromatography mass spectrometry (UHPLC-MS) untargeted metabolomics (Fig. 1).

With the demonstration of this workflow to discover rich and extensive information on the disposition of pharmaceuticals and related toxicodynamic knowledge presented in the following five results sections, here we describe the rationale for the computational workflow and selection of processing parameters. First, the workflow applies three intensity-based filters to the acquired MS[1] data, following peak picking, to refine the dataset towards a list of putative xenobiotic-related features: (1) Only features present in at least 80% of all xenobiotic-exposed biological samples are retained. In principle, xenobiotic-related features should be present in all samples taken from exposed biological system(s), however, considering the low concentrations of some analytes and routine occurrence of missing values in UHPLC-MS untargeted metabolomics datasets[33], some leniency is incorporated into the filter to reduce the false-negative rate. (2) Features present in more than 50% of biological control samples are removed. Although xenobiotic-related features should not be present in control samples, peak detection can occur in cases of system carry-over[34], or in the event of co-eluting peaks, hence leniency was introduced through this filter to reduce the false-negative rate. (3) Only features with ≥10-fold median intensity in exposed samples relative to control samples are retained (in principle the fold-change should be infinite for xenobiotic-related features). This threshold was derived based on an extensive review of toxicity datasets, which concluded that a ≥10-fold intensity change is highly unlikely for endogenous biochemicals (Supplementary Fig. 1). Subsequent exploratory correlation analysis of the putative xenobiotic-related features, i.e., those that pass these three filters, provides confidence that the features are closely related based on their co-responsiveness[35].

The list of putative xenobiotic-related features is then annotated and grouped to unveil features common to a single compound[36]. This includes consideration that features may arise from in-source modification, including adduct formation, heterodimerisation and fragmentation. The use of retention time data here enables the distinction between products of analytical and biological transformation, i.e., chromatographically separated features likely arise from unique compounds produced in biological test systems. Comparison of the grouped features to a list of predicted biotransformation products, generated by combining outputs of in silico prediction engines (e.g., SyGMa[37]) with previously reported experimental data, allows for putative compound annotation. Further confidence in these annotations can be gained from inspection of corresponding MS[2] fragmentation spectra. Since preparing analytical standards for each of the biotransformation products of >100,000 compounds[38,39] is impossible, the application of in silico fragmentation predictor tools (e.g., MetFrag[40]) is essential to attempt to assign partial or complete chemical structures to the measured MS[2] spectra. We use the structure corresponding to the putative compound annotation(s) and/or exposure xenobiotic structure as the input for these tools, since many xenobiotic biotransformation products will not be documented in publicly available databases. Annotation of a compound's MS[2]

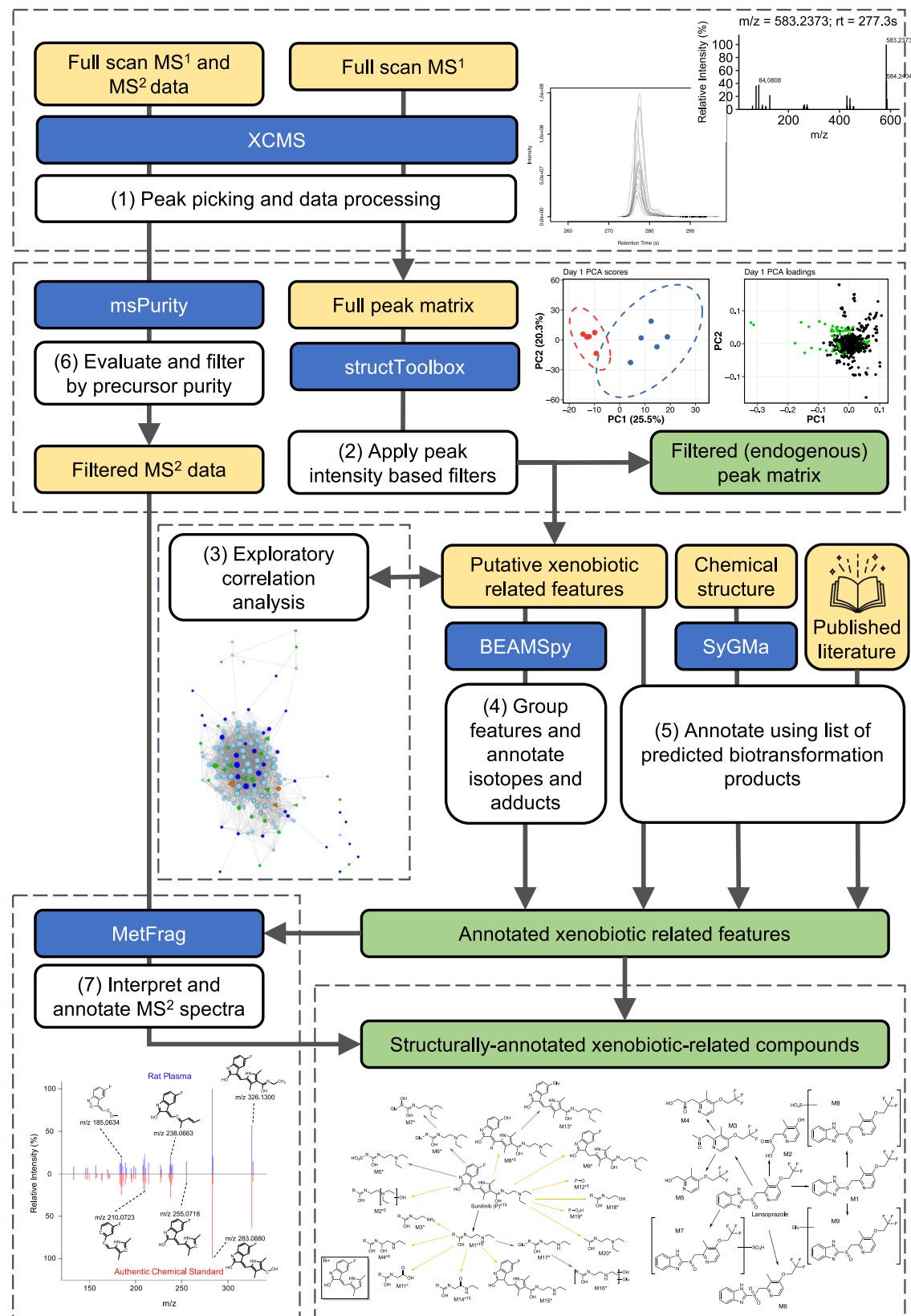

spectrum with substructures of the parent xenobiotic is consistent with it being related, via biotransformation, to the parent. The differences between the MS$^2$ spectra of the parent and suspected biotransformation product may elucidate the site of modifications[41,42].

The final output of the workflow is a list of detected exposure xenobiotic(s) and their biotransformation products, annotated with putative molecular formula, partial or complete chemical structures, and corresponding peak intensity data. These data can be further analysed to discover the fate of the xenobiotic, investigate associations between exposure and effects on endogenous biochemistry, and contrast the metabolic competency of biological test systems, as described below.

**Fig. 1 | Schematic of the untargeted ADME/TK workflow.** An open-source workflow to semi-automatically discover and visualise xenobiotics and their biotransformation products from UHPLC-MS untargeted metabolomics dataset. (1) Full scan MS[1] data are peak-picked and processed using XCMS[51] to produce a 'full peak matrix'. (2) A series of three peak intensity-based filters are applied using structToolbox[52] to reduce this matrix to generate a subset matrix of 'putative xenobiotic-related features' which can be explored initially by (3) correlation analysis to reveal relationships between features. (4) Grouping of features and annotation of peak patterns, followed by (5) annotation of compounds using a list of predicted biotransformation products, yields a matrix of annotated xenobiotic-related compounds and their relative intensities. Confidence in identifying the xenobiotics and their biotransformation products can be increased by incorporating MS[2] data measured for these annotated compounds. (6) MS[2] data are processed and filtered using msPurity[53] and (7) compared against in silico predicted assignments using MetFrag[40] to generate a list of 'structurally annotated xenobiotic-related compounds'. Green shading indicates the outputs of the workflow. Putative xenobiotic-related features are removed from the full peak matrix, generating a filtered (endogenous) peak matrix for endogenous data analysis.

## Untargeted workflow discovers cardiotoxin fate in rat

To test the ability of the workflow to discover pharmaceuticals and their biotransformation products from an untargeted metabolomics dataset, we applied it to an investigation of the metabolic perturbations induced in rats exposed to sunitinib (Supplementary Table 1). This dataset was originally measured to characterise endogenous metabolic changes only and comprised four assays, HILIC (Hydrophilic Interaction Liquid Chromatography) UHPLC-MS in positive and negative ion modes to study the polar metabolome, and reverse-phase (RP) $C_{18}$ UHPLC-MS in both ion modes to investigate the lipidome, for both plasma and cardiac samples. Application of the three intensity-based filters (Fig. 1) to 12,197 features in the full peak matrix from HILIC positive ion mode acquisition (HILIC-Pos) of plasma yielded 107 putative xenobiotic-related features. Similar filtering applied to the HILIC negative (HILIC-Neg), RP $C_{18}$ positive (RP-$C_{18}$-Pos) and RP $C_{18}$ negative (RP-$C_{18}$-Neg) datasets isolated 7, 30 and 2 features, respectively (Supplementary Table 2).

Correlation analysis of the putative sunitinib-related features (data from the four assays combined) discovered the majority of features are highly associated ($R \geq 0.75$, Pearson's), forming a dense cluster with representative features of sunitinib (Fig. 2a). The observation of this co-responsive behaviour provides further confidence that those features are sunitinib-related. Features peripheral to the core cluster are likely not related to sunitinib, or represent biotransformation products who's rate of synthesis is limited by other factors beyond the availability of sunitinib[35].

A list of predicted biotransformation products of sunitinib was generated from literature sources[30] and two in silico prediction tools, comprising 1492 unique molecular formulae (Supplementary Data 1, Supplementary Table 3). Of these, 109 were predicted by both prediction engines, 217 were uniquely predicted by SyGMa[37], and 1166 were predicted by only the 'Generate Expected Compounds' tool in Compound Discoverer (Thermo Scientific) (Fig. 2b). Comparison with the metabolomics data (Fig. 1) revealed 19 of the predicted biotransformation products, plus sunitinib, could be annotated in the HILIC-Pos dataset (Supplementary Fig. 2; Supplementary Table 3). Additionally, 2 and 5 products were annotated against the filtered HILIC-Neg and RP-$C_{18}$-Pos datasets, respectively, with some redundancy across datasets. No biotransformations were annotated within the RP-$C_{18}$-Neg dataset (Fig. 2c). Taken together, 19 predicted biotransformation products of sunitinib were detected and annotated, comprising 13 Phase I and 6 Phase II transformations (Fig. 2d). As such, the workflow successfully revealed all except one (M10, previously only reported in monkey faeces) of the 11 sunitinib biotransformations previously reported by ref. 30. (Supplementary Table 3). Of note, the workflow discovered a further 9 biotransformations of sunitinib not reported in the gold-standard pharmacokinetics study using [$^{14}$C]-radiolabelling[30] (M12–M20; Fig. 2b).

The identity of features annotated as the molecular ions ([M + H]$^{+}$/[M − H]$^{-}$) of sunitinib were confirmed by comparison against an authentic chemical standard. In silico structural annotation of the experimentally acquired MS[2] spectra of sunitinib (from analysis of an authentic chemical standard and exposed rat plasma) using MetFrag[40] revealed that peaks at $m/z$ 185, 210, 238, 255, 283 and 326 represent substructures of sunitinib (Fig. 2e). To increase confidence in the putative biotransformation product annotations derived from accurate mass matching, and to elucidate structural information, the experimentally acquired MS[2] spectra of features corresponding to the molecular ions of putative sunitinib biotransformation products were structurally annotated using MetFrag[40]. MS[2] spectra were acquired successfully for 12 of the 19 biotransformation products reported here (Supplementary Fig. 3). The acquired spectra confirmed all 12 compounds were sunitinib-related given the annotation of at least three of the substructures of sunitinib described above (Fig. 2e; Supplementary Data 2). Deviations of the MS[2] spectra of biotransformation products from that of sunitinib provided insights into the location of modifications, thus contributing to the structural elucidation of the previously unreported biotransformations. As such, we report the presence of sunitinib to Metabolomics Standards Initiative (MSI)[43] and Schymanski[44] confidence level 1 (using accurate $m/z$, retention time and MS[2] spectral match to authentic standard), 12 biotransformation products of sunitinib to MSI level 2 (Schymanski confidence level 2–3, i.e., probable or tentative structure, using accurate $m/z$ and in silico structural annotation of MS[2] spectra), and 7 biotransformation products with putative molecular formulae (Schymanski confidence level 4, unequivocal molecular formula, using accurate $m/z$), in the plasma of rats exposed to sunitinib (Fig. 2f; Supplementary Data 2). Eight of the biotransformation products detected included N-dealkylation of sunitinib (M1, M3, M4, M11, M14, M15, M16, M17) while half are the result of sunitinib oxygenation (M2, M4, M7, M8, M11, M12, M14, M16, M18, M19). M9 and M15 are products of saturation of the exocyclic double bond of sunitinib and M1, respectively. M20 results from the hydrolysis of the cyclic head from the tail. Considering the Phase II biotransformations detected, glucuronidation (M6, M7, M13, M16, M17) appears as the preferred conjugation reaction, while only a single sulphate conjugate (M5) was detected.

The workflow was also applied to data acquired from analysis of cardiac tissue collected from the same rats. In total, 68, 41, 137 and 53 putative sunitinib-related features were isolated from the HILIC-Pos, HILIC-Neg, RP-$C_{18}$-Pos and RP-$C_{18}$-Neg datasets, respectively (Supplementary Table 2). These lists included 6 predicted biotransformation products that were also detected in the plasma: M1, M2, M4, M6, M7 and M8 (Supplementary Data 2).

To further demonstrate the ability of the developed workflow to discover biotransformation products of an exposure pharmaceutical from an untargeted metabolomics dataset, we applied it to a second experiment designed to investigate the endogenous metabolic perturbations induced in rats exposed to KU60648 (Supplementary Table 1; Supplementary Note 1). The biotransformation map of this xenobiotic has not been reported previously. We discovered 22 biotransformation products of KU60648 in the plasma of rats; 4 of those products were also detected in the cardiac tissue of a subset of the same animals (Fig. 3, Supplementary Data 3).

## Temporally tracking pharmaceuticals and biotransformation products

In addition to providing qualitative data regarding the presence of a xenobiotic and its biotransformation products, untargeted metabolomics measurements also yield relative amounts of those compounds. For the experimental design reported here, this enables changes in

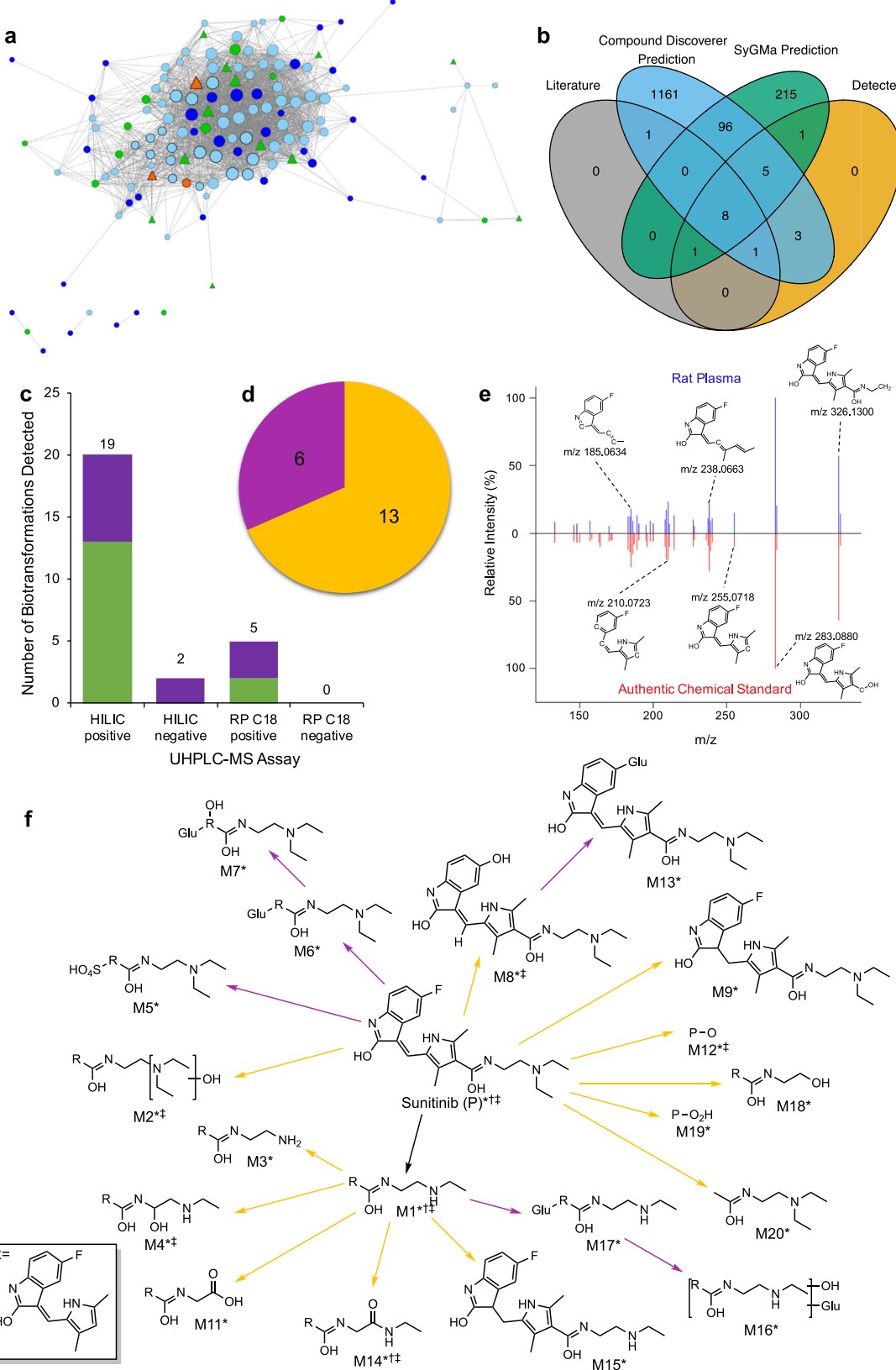

relative levels of each pharmaceutical and its biotransformation products in rat plasma to be tracked over time. Additionally, comparisons of relative amounts of the pharmaceuticals (and their biotransformation products) between individual test subjects allows judgement on the inter-individual variability in their disposition (Supplementary Table 1).

First, the reliability of the metabolomics UHPLC-MS peak intensity measurements of the parent substances, made whilst simultaneously measuring thousands of endogenous features, were verified by comparison to conventional quantification of the pharmaceuticals using single compound targeted liquid chromatography-tandem mass spectrometry (LC-MS/MS). A strong correlation of the intensities of

**Fig. 2 | Application of the untargeted ADME/TK workflow to plasma of rats exposed to sunitinib. a** Pearson's correlation-based network of putative sunitinib-related features (*n* = 29 biologically independent samples). Edges present when Holm adjusted *p* value < 0.05 and *R* ≥ 0.75, node size is proportional to its degree of connectivity. Node colour highlights molecular ions ($[M + H]^+$/$[M − H]^−$) of sunitinib (orange), molecular ions of biotransformation products of sunitinib (green), alternate ion forms (adducts and isotopes) of sunitinib or biotransformation products (pale blue), and unannotated features (dark blue). Node shape distinguishes between annotations based on $MS^1$ data only (circle) vs. more confident annotations also based on $MS^2$ data (triangle). The high density of this network is implicit of the expected strong correlations between features representing a single compound and chemically-related compounds. **b** The overlap of sunitinib biotransformation products reported in published literature[30], predicted by the 'Generate Expected Compounds' tool of Compound Discoverer (Thermo Scientific), and by SyGMa[37], and detected by untargeted metabolomics. **c** Number of

molecular formulae-annotated (putative annotation) (purple) and structurally-annotated (MSI level 2) (green) biotransformation products of sunitinib detected by each UHPLC-MS metabolomics assay. The bars are annotated with the total number of biotransformation products detected by each UHPLC-MS metabolomics assay. **d** The proportion of Phase I (yellow) and Phase II (magenta) biotransformation products of sunitinib detected across all assays. **e** Representative comparison of measured $MS^2$ fragmentation spectra for sunitinib in rat plasma (top) vs authentic chemical standard of sunitinib (bottom) and the corresponding MetFrag-annotated structures of major peaks. **f** A biotransformation map of sunitinib showing the biotransformation products discovered in the rat plasma UHPLC-MS untargeted metabolomics dataset (data from four assays – HILIC positive (*)/negative (†), RP C_18 positive (‡)/negative (§)) by the untargeted ADME/TK workflow. The colour of the arrow denotes type of transformation– Phase I (yellow) or Phase II (magenta). Extracted ion chromatograms and $MS^2$ fragmentation spectra for these compounds are displayed in Supplementary Figs. 2 and 3.

sunitinib measured in the HILIC-Pos metabolomics assay with those from the fully quantitative approach, for plasma samples collected on day 15 (samples for metabolomics taken 4 h after targeted measurements, from the same individuals), was observed (*R* = 0.90; Fig. 4a). High correlation coefficients were also calculated for the HILIC-Neg and C_18-RP-Pos assays relative to the conventional targeted approach (Supplementary Fig. 6). Comparable analysis using intensity measurements from untargeted metabolomics, and quantitative measurements from targeted LC-MS/MS of KU60648 provided further evidence of the reliability of intensity data from untargeted metabolomics (Supplementary Fig. 7).

The metabolomics measurements—originally intended to investigate the endogenous biochemical changes over time—were used to track the relative temporal changes in sunitinib systemic exposure over the 15-day study (Fig. 4b). Sunitinib intensities increase towards steady state with repeated daily dosing between days 1 and 4. At day 15, 28 h after the last dose on day 14, average levels of sunitinib are significantly two-fold less than those measured 4 h post-dose on day 8, providing evidence of effective elimination upon completion of dosing. Comparison of relative sunitinib levels across individual rats revealed greatest variability at the points of greatest systemic exposure (days 4 and 8; Fig. 4b).

Similar analyses were conducted for KU60648 over the course of the 4-day rat exposure study (Supplementary Fig. 8). Significantly greater systemic exposure of KU60648 is achieved after a second dose. Meanwhile, levels measured in the plasma on day 4 indicate some elimination by 24 h post-dose.

The untargeted ADME/TK workflow also provides peak intensity measurements for the discovered biotransformation products, representing additional information that is not measured in conventional toxicokinetic studies. K-means clustering of the temporal responses of sunitinib and its biotransformation products (Fig. 2f) discovered commonality between the temporal variation in the systemic exposure of sunitinib and the responses of the products of its simplest routes of metabolism: M1 (de-ethylation), M2 (hydroxylation) and M9 (saturation) (Figs. 2f; cluster 2, 4c). These compounds display significantly greater systemic exposure on days 2, 4 and 8 compared with day 1. A significant drop in levels is observed on day 15 to levels comparable to day 1, evidencing effective elimination once dosing is ceased. Five other clusters of biotransformation products diverge from the behaviour of the parent xenobiotic. Cluster 1 (M11, M15, M17 and M19) and cluster 6 (M3, M4, M6, M12, M14 and M18) show similar trends to cluster 2, with a significant increase from day 1 levels observed on days 4 and 8, and just day 8, respectively. The majority of detected biotransformation products resulting from Phase II conjugation reactions, excluding M6 and M17, do not show any significant changes in systemic levels across the measured time points (clusters 3, 4 and 5). M8 (product of oxidative de-fluorination) and M20 (product of parent cleavage), also in cluster 5, displayed similar trends (Figs. 2f, 4c). This

suggests the steady state of these compounds is reached following the initial dose.

Temporal responses of the biotransformation products of KU60648 discovered in the plasma of exposed rats were also investigated. K-means clustering revealed six clusters of responses (Supplementary Fig. 9). The responses of M4, M6, M9, M11, M18 and M19 were found to be most similar to that of KU60648, forming cluster 2, while five other clusters of biotransformation products diverge to varying degrees from the behaviour of the parent.

## Lipid responses associate with internal dose

The capability of the untargeted ADME/TK workflow to reveal the identities and relative levels of xenobiotic-related features from an untargeted metabolomics dataset may be beneficial to the analysis of the endogenous responses. First, by accurately removing the xenobiotic-related features from the metabolomics dataset, a more reliable multivariate statistical analysis of the endogenous data can be achieved. This is demonstrated by principal component analyses (PCA) (Fig. 5a, b; Supplementary Fig. 10). Where putative sunitinib-related features (Supplementary Table 2) are not removed from the metabolomics dataset, these features are primarily responsible for the separation of exposed plasma samples from biological controls along PC1, as evidenced by the loadings plots. This influence of exposure substance-related features leads to the false conclusion of a strong perturbation of the plasma metabolome in response to sunitinib on day 1 (Fig. 5a). At day 15, although sunitinib-related features have the highest magnitude loadings, separation of sample classes remains along PC1 after their removal, indicating a strong perturbation of the endogenous fraction of the detected plasma metabolome as a consequence of sunitinib exposure (Fig. 5b). Similar analysis of day 2, day 4 and day 8 measurements reveals a trend of increasing perturbation over time, which is only discernible when sunitinib-related features are excluded (Supplementary Fig. 10).

This trend of an increasing magnitude of perturbation is also revealed by univariate statistical analyses conducted on the filtered (endogenous) peak matrices (Supplementary Table 2). Specifically, a total of 40, 121, 131, 463 and 3494 features were significantly perturbed across the four assays on day 1, 2, 4, 8 and 15, respectively (Supplementary Table 4). Changes are observed across a broad array of metabolite and lipid classes (Supplementary Data 4).

PCA and univariate analyses conducted on the cardiac filtered (endogenous) peak matrices (Supplementary Table 2) indicate a distinct perturbation of the cardiac metabolome by sunitinib (Supplementary Fig. 11). A total of 455 features were significantly perturbed across the four assays (Supplementary Table 5) with effects on the lipidome dominating (Supplementary Data 5).

Inter-individual differences in internal dose should in principle relate strongly to the magnitude of change of the endogenous metabolome, thereby offering an effective route for discovering xenobiotic-

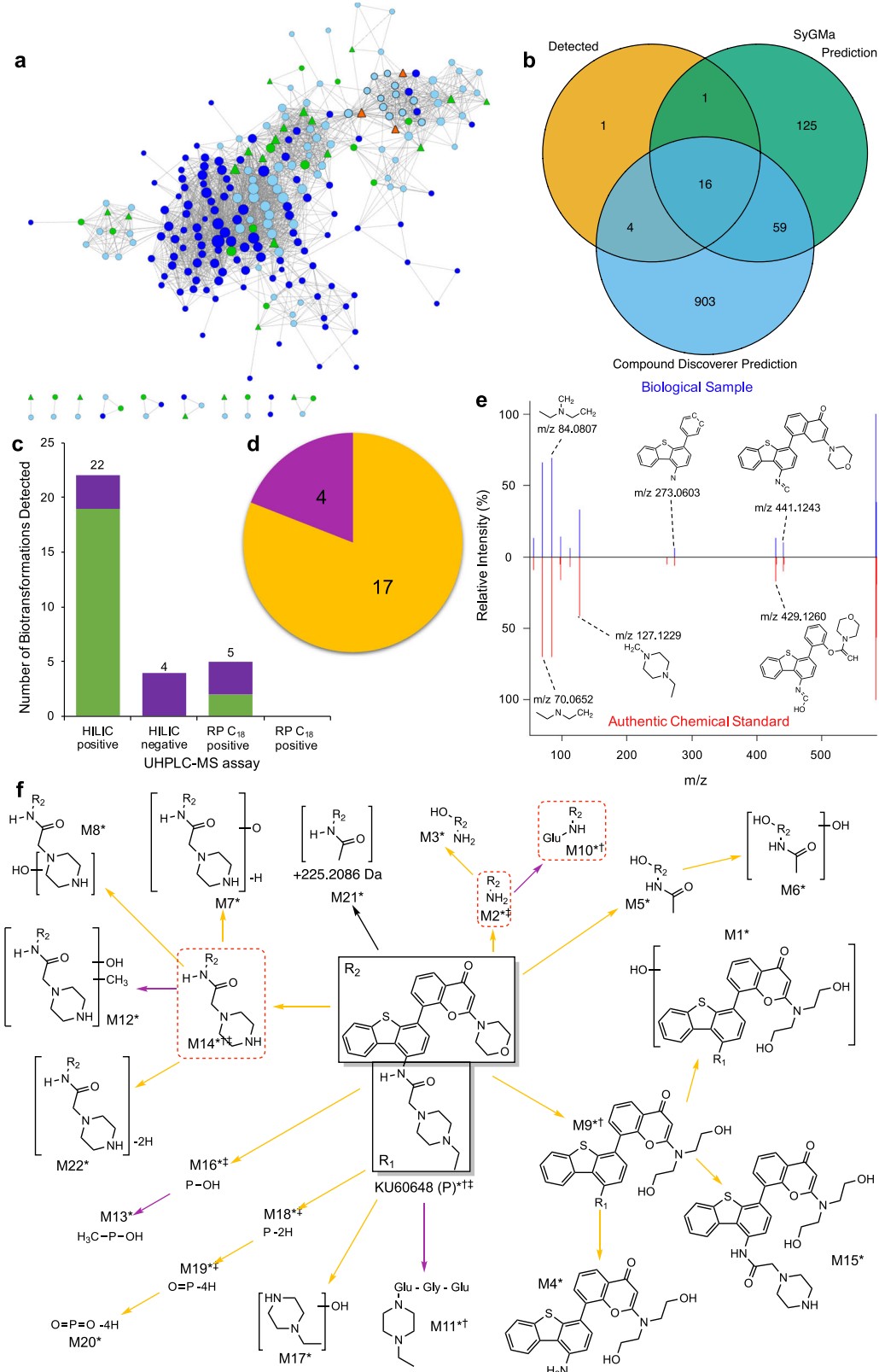

induced perturbations. To this end, measurements of the relative levels of pharmaceutical were used to explore the endogenous responses, i.e., to attempt to discover direct associations between exposure substance levels at a target site and endogenous lipids. This would not be possible with nominal dose values which do not account for inter-individual differences in bioavailability and distribution. Specifically, correlation analysis revealed a dense network of direct

associations between sunitinib levels and subsets of multiple lipid classes in the cardiac tissue of exposed rats (Fig. 5c; Supplementary Data 6).

Nine sphinogomyelins negatively correlated with the relative internal levels of sunitinib; over-representation analysis confirming that this lipid class is significantly overrepresented ($p < 0.1$) amongst those lipids whose responses correlate with levels of sunitinib

**Fig. 3 | Application of the untargeted ADME/TK workflow to plasma of rats exposed to KU60648. a** Pearson's correlation-based network of putative KU60648-related features ($n = 19$ biologically independent samples). Edges present where Holm adjusted $p$ value $< 0.05$ and $R \geq 0.75$, size of node is proportional to its degree connectivity. Node colour highlights molecular ions ($[M + H]^+/[M − H]^−$) of KU60648 (orange), molecular ions of biotransformation products (green), alternate ion forms (isotopes and adducts) of KU60648 or biotransformation products (pale blue), and unannotated features (dark blue). Node shape distinguishes between annotations based MS[1] data only (circle) vs annotations using MS[2] data also (triangle). The high density of this network is implicit of the expected strong correlation between features representing a single compound and chemically-related compounds. **b** The overlap of KU60648 biotransformation products predicted by the 'Generate Expected Compounds' tool of Compound Discoverer (Thermo Scientific), and by SyGMa[37], and detected by untargeted metabolomics. **c** Number of molecular formulae-annotated (putative annotation) (purple) and structurally-annotated (MSI level 2) (green) biotransformation products of

KU60648 detected by each UHPLC-MS metabolomics assay. The bars are annotated with the total number of biotransformation products detected by each UHPLC-MS metabolomics assay. **d** The proportion of Phase I (yellow) and Phase II (magenta) biotransformation products of KU60648 detected across all assays. **e** Representative comparison of measured MS[2] fragmentation spectra for KU60648 in rat plasma (top) vs authentic chemical standard of KU60648 (bottom) and the corresponding MetFrag-annotated structures of major peaks. **f** A biotransformation map of KU60648 showing biotransformation products discovered in the rat plasma UHPLC-MS untargeted metabolomics dataset (data from four assays – HILIC positive (*)/negative (†), RP C$_{18}$ positive (‡)/negative (§)) by the untargeted workflow. The colour of the arrow denotes type of transformation– Phase I (yellow) or Phase II (magenta). Outlined (red dashed line) are the biotransformation products also detected in the cardiac tissue of exposed rats. Extracted ion chromatograms and MS[2] fragmentation spectra for these compounds are displayed in Supplementary Figs. 4 and 5.

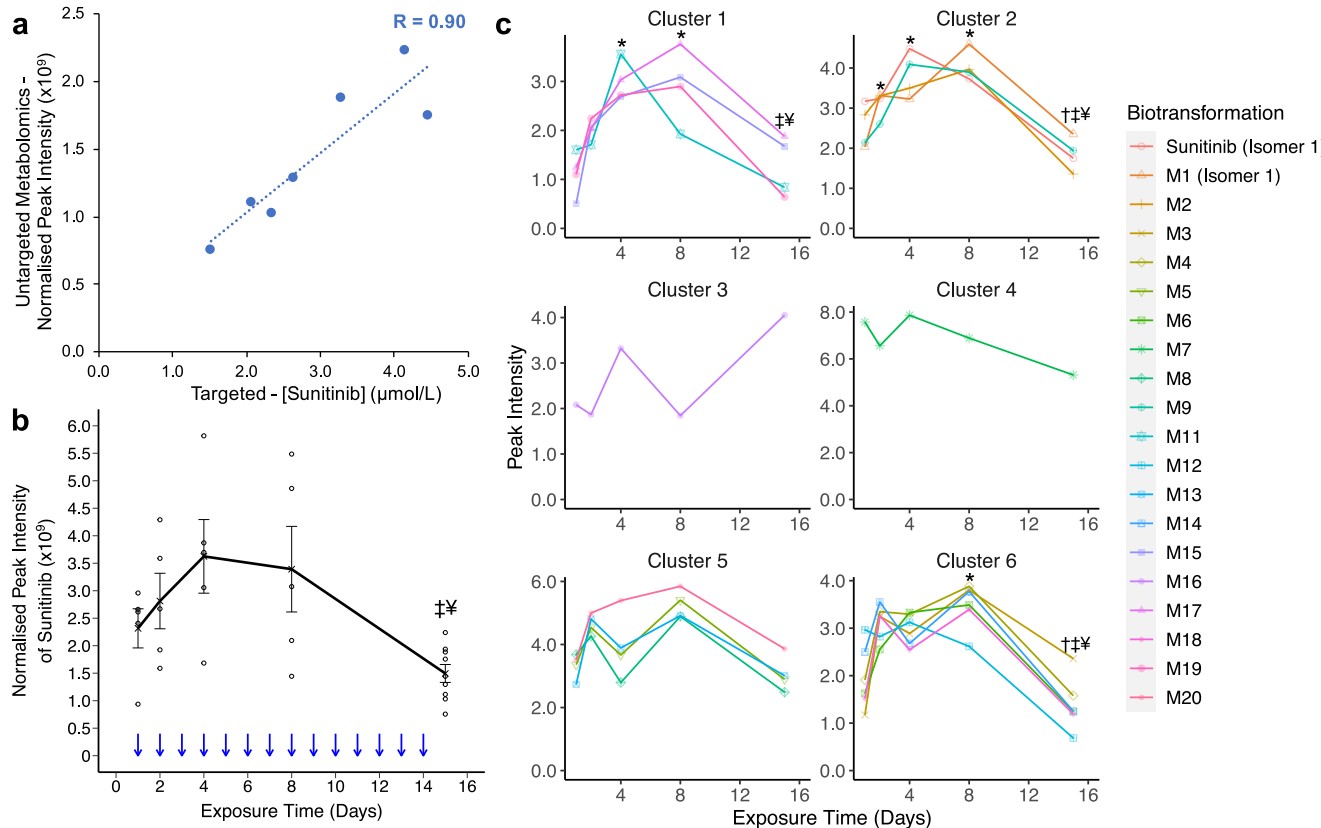

**Fig. 4 | Peak intensity measurements of sunitinib and its biotransformation products in rat plasma. a** Relationship between UHPLC-MS untargeted metabolomics peak intensity measurements of sunitinib (HILIC positive assay) and the absolute quantification of sunitinib using conventional targeted LC-MS/MS. **b** Mean peak intensity (cross) of sunitinib over the duration of the 15-day study, as measured by UHPLC-MS untargeted metabolomics. Individual data points are also displayed (open circle). Error bars show standard error. Arrows indicate time of dosing. **c** Median peak intensities, measured by UHPLC-MS untargeted metabolomics and scaled by unit variance, of sunitinib and its biotransformation products over the duration of the 15-day exposure study, clustered by an unsupervised k-means approach ($k = 6$, optimal value determined by the Elbow Method). Cluster 1: M11, M15, M17, and M19. Cluster 2: sunitinib, M1, M2 and M9. Cluster 3: M16.

Cluster 4: M7. Cluster 5: M5, M8, M13 and M20. Cluster 6: M3, M4, M6, M12, M14, and M18. Statistical analysis (**b**, **c**) was conducted by one-way ANOVA followed by Tukey's post-hoc test. Significance is displayed as follows: * $p < 0.05$ vs. day 1, † $p < 0.05$ vs. day 2, ‡ $p < 0.05$ vs. day 4, ¥ $p < 0.05$ vs. day 8). Specifically, **b** $p = 0.019$ (day 15 vs. day 4) and $p = 0.0435$ (day 15 vs. day 8); **c**, cluster 1: $p = 0.0001$ (day 4 vs day 1), $p = 0.0010$ (day 8 vs. day 1), $p = 0.0046$ (day 15 vs. day 4) and $p = 0.0382$ (day 15 vs. day 8); cluster 2: $p = 0.0491$ (day 2 vs. day 1), $p = 0.0001$ (day 4 vs. day 1), $p \leq 0.0001$ (day 8 vs. day 1), $p = 0.0015$ (day 15 vs. day 2), $p \leq 0.0001$ (day 15 vs. day 4 and day 15 vs. day 8); cluster 6: $p = 0.0001$ (day 8 vs. day 1), $p = 0.0001$ (day 15 vs. day 2), $p = 0.0001$ (day 15 vs. day 4), $p < 0.0001$ (day 15 vs. day 8). $N = 5$ individual animals on days 1, 2, 4, and 8; $n = 9$ individual animals on day 15. Source data for this figure are provided in the Source Data file.

(Supplementary Table 6). This finding is supported by the conventional statistical analyses of the endogenous cardiac data for which seven sphingomyelins decreased significantly upon exposure, with a mean fold-change of 0.71 (Supplementary Data 6). Related, levels of seven sphingomyelins were significantly higher in the plasma of

exposed rats compared to biological controls on day 15 (Supplementary Data 5). Of note, four of those overlap with the nine that negatively correlated to sunitinib levels at the site of toxicity (Fig. 5d). Taken together, these analyses indicate sphingomyelins play a role in the progression of sunitinib-induced cardiotoxicity, given their significant

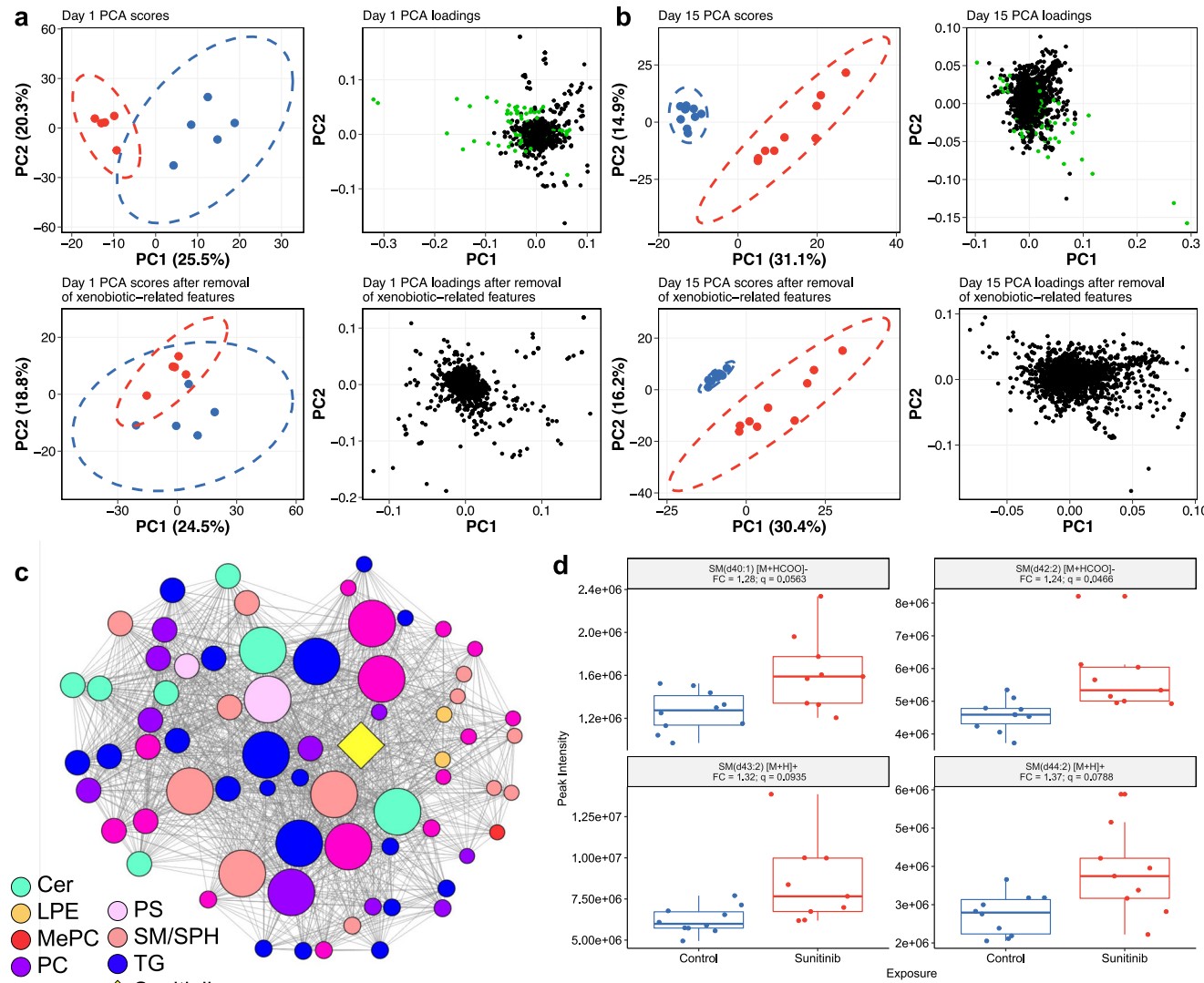

**Fig. 5 | Biochemical changes revealed in the plasma and cardiac tissue of rats exposed to sunitinib.** PCA scores and loadings plots of plasma samples from rats exposed to sunitinib (red) for (**a**) 1 and (**b**) 15 days and time-matched biological control rats (blue) measured by HILIC UHPLC-MS in positive ion mode before and after removal of putative sunitinib-related features (green in loadings plots).
**c** Correlation network of sunitinib and MSI level 2 annotated lipids: ceramides (cer), lysophosphatidylethanolamines (LPE), methyl phosphatidylcholine (MePC), phosphatidylcholines (PC), phosphatidylethanolamines (PE), phosphatidylserines (PS), sphingomyelins (SM/SPH) and triacylglycerols (TG), in cardiac tissue of rats exposed to sunitinib (*N* = 5). Node size is proportional to its degree of connectivity.

Edges represent significant Spearman's correlation ($p < 0.05$ and, $|\rho| \geq 0.9$) between compounds. All nodes displayed are significantly correlated to sunitinib. **d** Box plots showing significantly increased plasma levels of four sphingomyelins found to be associated with sunitinib levels at the site of toxicity, in rats exposed to sunitinib (red) for 15 days, compared to time-matched biological controls (blue). Boxes show the interquartile range (IQR), with the line representing the median, and the whiskers showing 1.5× IQR. Data is from *n* = 9 individual animals. Fold change and *q* values (FDR-corrected *p* values calculated by Student's two-tailed *t*-test) are displayed. Source data for (**d**) are provided in the Source Data file.

perturbation and direct association with sunitinib levels within the cardiac tissue, and are amenable to use as biomarkers for sunitinib-induced cardiotoxicity, given the measurable effects on their circulating levels.

A significant perturbation of the plasma and cardiac metabolome of rats exposed to KU60648 was also observed following removal of putative KU60648-related features (Supplementary Table 2; Supplementary Figs. 12, 13; Supplementary Table 7). In the plasma, a total of 5771 and 5696 features were significantly perturbed across the four assays on days 2 and 4, respectively, indicating extreme effects of KU60648 exposure (Supplementary Table 7 and Supplementary Data 7). At the site of toxicity (cardiac tissue), perturbation of lipids dominated the response to KU60648 exposure (Supplementary Table 7 and Supplementary Data 8). A correlation-based approach was used to seek associations between

levels of KU60648 in the cardiac tissues of exposed rats and the responses of MSI level 2 annotated lipids in the same samples (Supplementary Fig. 14). Significant correlations were observed, mostly for acylcarnitines, phosphatidylcholines and triacylglycerols (Supplementary Data 9). Direct association of these lipids to KU60648 levels explains the inter-individual variation of their relative levels amongst exposed samples.

Over-representation analysis identified the negative correlation of a subset of eight acylcarnitines to relative amounts of KU60648 within the cardiac tissue of exposed rats as significant (Supplementary Table 8). Of note, conventional univariate analysis (*t*-test) of this dataset did not reveal acylcarnitines in the cardiac tissue of exposed rats as significantly different compared to controls. This is likely due to the large inter-individual variation amongst exposed samples. The explicit coupling of individual- and site-specific KU60648 levels

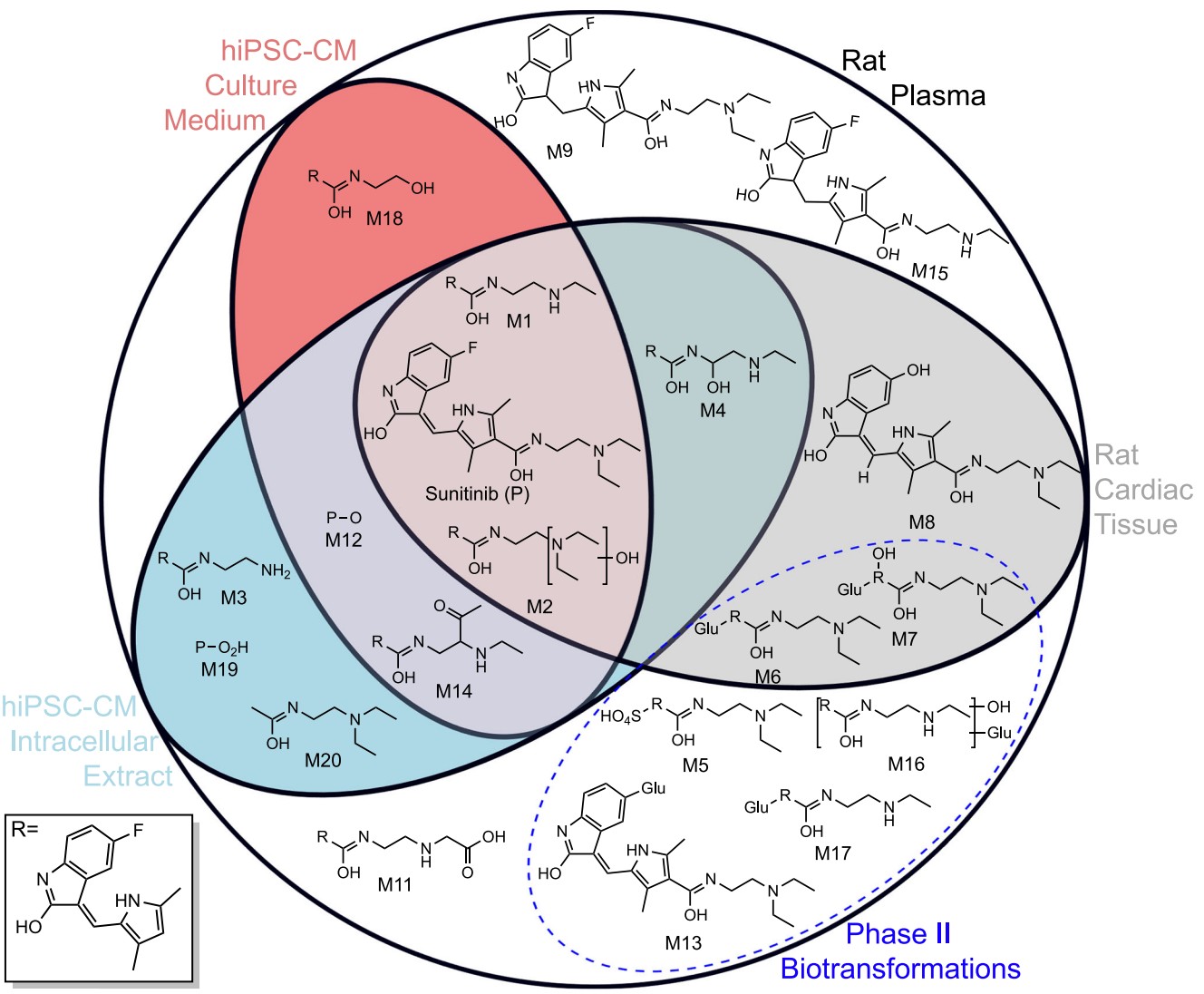

**Fig. 6 | Comparison of the metabolic competencies of cardiomyocyte (hiPSC-CM) in vitro cultures and rat by applying the untargeted ADME/TK workflow to four UHPLC-MS metabolomics datasets.** Euler diagram depicting in which biological samples (rat plasma, rat cardiac tissue, the intracellular extracts and culture medium of hiPSC-CM cultures) sunitinib and its biotransformation products were detected.

(i.e., internal dose) with measurements of the lipidome was therefore required to reveal that the response of acylcarnitines is highly sensitive to the relative exposure of cardiac tissue to KU60648, and thereby to discover the involvement of acylcarnitines in KU60648-induced cardiotoxicity. Levels of five of these acylcarnitines, alongside four others, were further found to be significantly increased in the plasma of exposed rats on days 2 and 4, with median fold changes of 7.80 and 12.94, respectively (Supplementary Data 7; Supplementary Fig. 15). The evidenced direct association with KU60648 levels at the site of toxicity, coupled with large and significant disruption in the plasma, makes these molecules attractive putative biomarkers for the onset of drug-induced cardiotoxicity.

**Contrasting metabolic competencies of experimental models**

In vitro models for assessing organ toxicities are becoming increasingly common in accordance with the 3Rs concept (reduction, refinement, and replacement of animal testing)[1,45,46] and the advantages of human-relevant test systems[31]. Xenobiotic biotransformations can play a major role in toxicity, however many in vitro model systems remain uncharacterised in terms of their metabolic competencies[32,47]. Here we applied the untargeted ADME/TK workflow to metabolomics datasets that were originally collected to investigate the effects of

sunitinib on hiPSC-CMs. By comparison to biotransformation of sunitinib in rat, we demonstrate the ability to reveal the metabolic competences of experimental model in vitro systems.

Untargeted metabolomics measurements of both intracellular extracts and culture medium of hiPSC-CM cultures were acquired by the same four assays and processed using the workflow (Fig. 1). A total of 356 and 87 putative sunitinib-related features were identified in intracellular extracts and culture medium, respectively (Supplementary Table 9). This included sunitinib and its biotransformation products M1, M2, M3, M4, M12, M14 and M20 within the intracellular extracts, and sunitinib, M1, M2, M12 M14 and M18 in the culture medium (Fig. 6a; Supplementary Data 10). The observation of products of sunitinib de-ethylation, oxidation and/or hydrolysis demonstrates that cardiomyocytes possess metabolic competency, yet this appears to be limited to a subset of Phase I biotransformations. Meanwhile, investigation of the temporal distribution of sunitinib levels provides evidence, through lack of significant change, that hiPSC-CMs were exposed to a consistent level of sunitinib over the duration of the 24 h experiment (Supplementary Fig. 16).

Some overlap in the biotransformation products detected in cardiomyocyte cultures and cardiac tissue of rats was observed, mainly M1, M2 and M4. However, three products were unique to cardiac

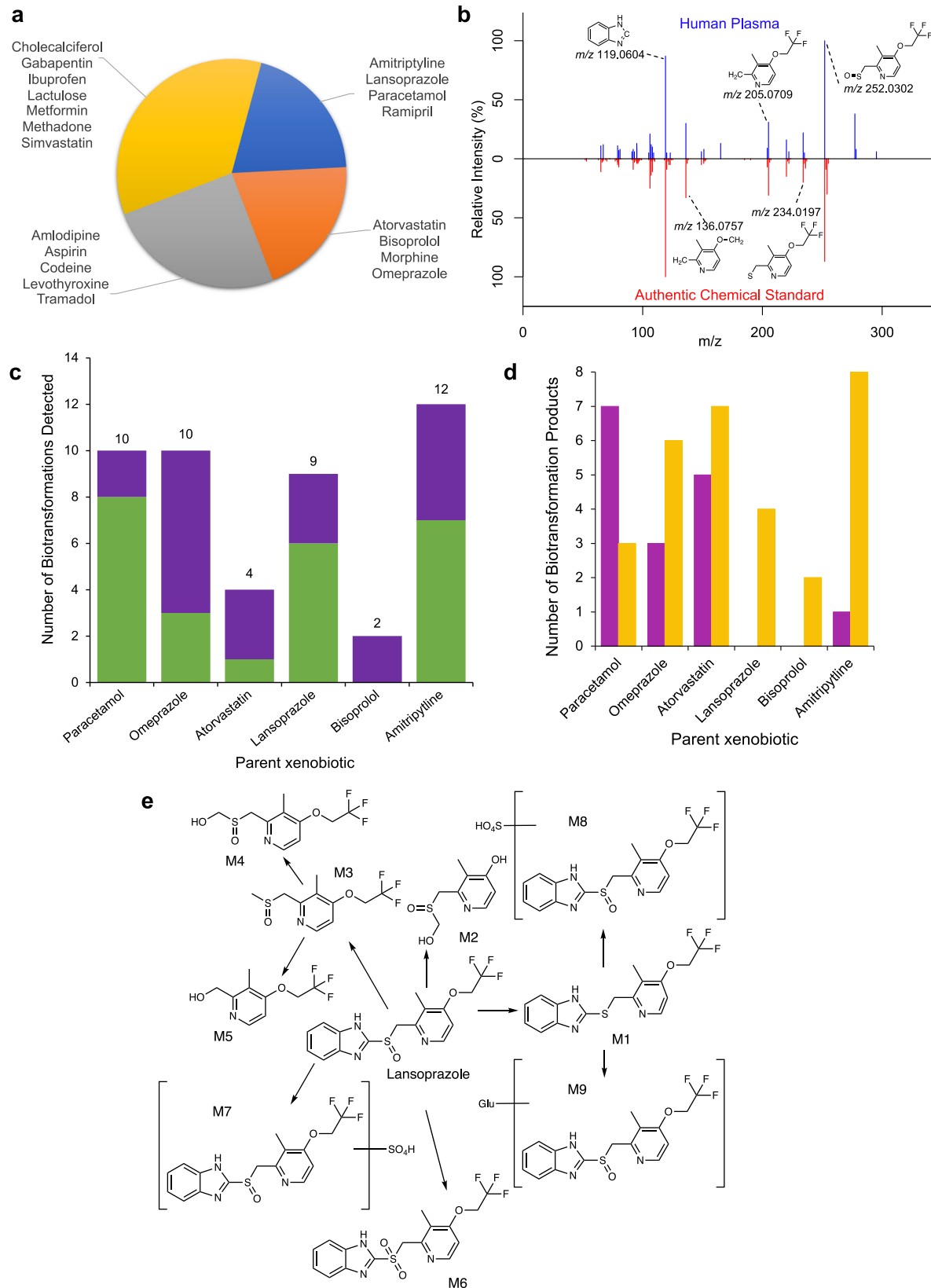

tissue, including two glucuronidation biotransformations (M6 and M7). Five biotransformation products were detected in the cardiomyocytes but not the cardiac tissue. Although this comparison may be confounded by differences in detection limits between matrices, it has revealed distinct patterns in the biotransformations of the pharmaceutical.

## Untargeted workflow discovers fate of pharmaceuticals in humans

The ability to characterise the exposure to, and biotransformation of, xenobiotics in humans as a component of an untargeted metabolomics assay of biochemical effects could enhance mechanistic safety assessments of pharmaceuticals, biocides and industrial chemicals.

**Fig. 7 | Discovery of pharmaceuticals and their biotransformation products in a human plasma untargeted metabolomics dataset. a** Pie chart displaying detection of UK National Health Service (NHS) top 20 most prescribed pharmaceuticals within the human plasma UHPLC-MS untargeted metabolomics dataset: blue—pharmaceuticals that were detected, with identity confirmed by matching retention time and MS[2] to analytical standard, orange—pharmaceuticals detected, with confidence in annotation from matching MS[2] to public database, yellow—pharmaceuticals detected according to putative MS[1]-based annotation, grey—not detected. **b** Representative comparison of measured MS[2] fragmentation spectrum for lansoprazole in human plasma (top) vs. its authentic chemical standard (bottom) and the corresponding MetFrag-annotated structures of major peaks. **c** Number of molecular formulae-annotated (putative annotation) (purple) and structurally-annotated (MSI level 2) (green) biotransformation products of detected and confidently annotated parent pharmaceuticals. **d** Number of Phase I (yellow) and Phase II (magenta) biotransformation products derived from six confidently annotated parent pharmaceuticals discovered by the workflow. **e** Map showing the biotransformation products of lansoprazole discovered in plasma of healthy humans by UHPLC-MS untargeted metabolomics using our untargeted ADME/TK workflow.

Additionally, characterising unknown and/or unintended human exposure to xenobiotics is important in environmental health research, e.g., to identify populations exposed through the environment to xenobiotics of concern. Here we demonstrate the capability of our workflow to detect human exposure to pharmaceuticals, from a list of suspects, and discover their metabolic fates (Supplementary Table 1; Fig. 7).

A list of the UK National Health Service (NHS) top 20 most prescribed pharmaceuticals was selected as an appropriate 'suspect' list for demonstrating this application of the workflow (Supplementary Table 10). All 20 of the suspects are considered emerging chemicals of concern and listed within the NORMAN[48] and CECscreen[49] databases. Eight of these pharmaceuticals were confidently detected (MSI level 1 or 2, or Schymanski confidence level 1–3) within the human plasma untargeted metabolomics dataset (Fig. 7a, b Supplementary Fig. 17; Supplementary Table 10). Subsequent labelling of each sample as either 'exposed' or 'control' for each of the eight confidently detected pharmaceuticals, followed by application of the three intensity-based filters (Fig. 1) generated matrices of putative xenobiotic-related features for six parent substances (each detected in at least two human plasma samples). Annotation of these features using BEAMSpy[50] and SyGMA[37] predictions revealed 10, 10, 4, 9, 2 and 12 biotransformation products of paracetamol (acetaminophen), omeprazole, atorvastatin, lansoprazole, bisoprolol, and amitriptyline, respectively (Fig. 7c; Supplementary Table 11; Supplementary Data 11–16; Supplementary Fig. 18). Of the total of 47 biotransformation products detected, 31 are products of Phase I transformations, while 16 result from Phase II transformations, mainly glucuronidation (Fig. 7d). Corresponding MS[2] spectra for 53% of the biotransformation products were measured and interpreted, providing confidence in their annotation (Fig. 7c, e; Supplementary Fig. 19; Supplementary Fig. 20; Supplementary Table 11; Supplementary Data 11–16).

## Discussion

The ability to measure xenobiotics and their biotransformation products in the same UHPLC-MS analysis used to probe endogenous biochemical effects enables insights into both the external chemical environments to which individuals are exposed and the internal fate and effects of xenobiotics, without the need for additional samples and analytical measurements, and hence associated costs. We implemented a data processing workflow to semi-automatically discover measurements corresponding to pharmaceuticals and their biotransformation products within untargeted UHPLC-MS metabolomics datasets, centred on three principles: (1) unbiased data-driven discovery of unknown biotransformation products. This is facilitated by exploiting the high sensitivity of high resolution, high mass accuracy mass spectrometry, enabling discovery of lower abundance molecules, and the application of three intensity-based filters to untargeted metabolomics datasets, which were designed to reduce the frequency of false-positive (fold-change filter) and false-negative (exposed and biological control sample filters) outputs, and requires no prior knowledge of xenobiotic biotransformations. Unlike other strategies for measuring xenobiotic biotransformations, this approach does not require stable isotope-labelling, opening the door to discovering unknown exposures to xenobiotics and their internal fate. (2) Targeted detection of known and/or predicted biotransformation products.

Primarily achieved through the application of in silico biotransformation prediction engines, this aids annotation of xenobiotic biotransformation products; e.g., combining SyGMa[37] and the 'Generate Expected Compounds' tool in Compound Discoverer, we successfully predicted 90% of the observed biotransformation products of sunitinib and KU60648. (3) Maximising confidence in the identity of detected xenobiotics and biotransformation products, largely through MS[2]-based structural elucidation; e.g., this approach was used to confirm identities of 26 of the 41 detected sunitinib and KU60648 biotransformation products. In the absence of MS[2] fragmentation data, e.g., for low intensity or low purity precursor ions, the combination of an accurate mass match to a predicted biotransformation product ('targeted') along with passing filtering thresholds ('data driven'), provides substantial evidence that compounds are xenobiotic-related. Taken together, the untargeted ADME/TK workflow can discover predicted and novel biotransformation products of unlabelled pharmaceuticals as well as provide confidence in their annotation or identity. The use of open-source software[37,40,51–53] following application of published UHPLC-MS-based untargeted metabolomics analytical methods[54] makes this approach amenable to application by the wider metabolomics community.

Applying this workflow to data collected from plasma samples of exposed rats, we discovered both previously reported and unreported biotransformation products for two cardiotoxins, sunitinib and KU60648. For the former, consistent with previous findings[30,55], we detected products of de-ethylation, oxidation, dehydrogenation, oxidative defluorination, and glucuronide and sulfate conjugation. The workflow additionally discovered biotransformation products resulting from combinations of these reactions in rat plasma (M12–M20; Fig. 2). Hence, our approach has supported and furthered prior knowledge on the biotransformation of sunitinib in rats using a dataset primarily intended to provide insights into endogenous metabolic responses. The discovery of biotransformation products of sunitinib was enabled through the use of high resolution, high mass accuracy mass spectrometry which offers enhanced sensitivity over [14C]-labelling-based approaches. We also report on the biotransformation of KU60648; we measured products of de-ethylation, oxidation, dehydrogenation, methylation and glucuronide and glycine conjugations. From similar analyses of cardiac tissue from the same rats, we gained insights into the distribution of sunitinib, KU60648 and their biotransformation products to a site of toxicity, the heart. The detection of sunitinib in cardiac tissue of rats builds on previous findings that demonstrated distribution of sunitinib-related material to the heart following dosing with [14C]-sunitinib[30]. The distribution of sunitinib, M1, M2 and M9 to the liver, kidney and tumour tissue has been reported in syngeneic subcutaneous mouse tumour models by MALDI-MS[56]; the application of our workflow has furthered the knowledge on the distribution of sunitinib biotransformation products by discovering the presence of Phase I (M1, M2, M4, M8—de-ethylation, oxidation, oxidative defluorination) and Phase II (M6, M7—glucuronidation) biotransformation products in rat heart. Similarly, we detected KU60648 and three of its biotransformation products (M2, M10 and M14—hydrolysis, glucuronidation and de-ethylation) in cardiac tissue of KU60648-exposed rats. The capability to examine the distribution of a xenobiotic and its biotransformation products to a site of toxicity could, for example, improve the ability to

decipher the causative agent of toxicity by providing compound-specific evidence of tissue exposure.

Analysis of the relative peak intensity measurements afforded by the metabolomics assays provided evidence of increases in systemic exposure with daily dosing, towards a steady state, and systemic clearance within 24 h of the final dose, of both sunitinib and KU60648. Time-resolved measurements of relative systemic exposure of a xenobiotic made by the same analysis, within the same samples, as the endogenous response, could, by association, help to explain the temporal trends detected in toxicological response(s) of the test subjects. Thus, through application of our workflow, untargeted metabolomics offers an effective route for discovering dose-response associations. Furthermore, analysis of the time-resolved responses of the biotransformation products revealed clusters of co-responsiveness. Such analysis of temporal changes in relative systemic levels of biotransformation products is not usually investigated. However, it provides potentially important insight into the fate of biotransformation products, some of which may possess efficacious or toxic activities, which could aid explanation of temporal toxicological responses.

The biotransformation capacity of in vitro models are typically not characterised and often assumed to be low. We reveal, through application of our workflow, that hiPSC-CMs have a previously unreported capacity for Phase I-type oxidative and hydrolytic reactions, but no indications of Phase II reactions. Such discoveries of in vitro biotransformation, by comparison to in vivo biotransformation, could help to explain discrepancies between in vitro and in vivo toxicological responses to a xenobiotic[31], without requiring additional time-consuming and expensive analysis.

In addition to the discoveries on the fate of pharmaceuticals, we mined the metabolomics dataset to investigate the endogenous biochemical responses of rats, in both plasma and at a site of toxicity (cardiac tissue), following exposure to sunitinib and KU60648. First, as demonstrated through PCA, xenobiotic-related features can be considered confounding factors in the analysis of endogenous biochemical responses so must be excluded from the dataset prior to analysis. Once removed, a strong endogenous biochemical response to exposure was revealed in plasma (at later time points) and cardiac tissue. Conventionally applied univariate analysis uncovered a number of compounds which significantly responded to exposure, predominantly various classes of lipids including ceramides, phospholipids, sphingolipids and di- and tri-acylglycerols in the cardiac tissue. This is consistent with the known association of perturbed lipid uptake and metabolism with cardiomyopathy[57,58]. Going beyond conventional analysis, we utilised the availability of relative internal dose and untargeted metabolomics measurements from the same samples to discover dose-response associations within the cardiac tissue. We discovered that the responses of sphingomyelins were strongly negatively correlated to the level of sunitinib in heart, providing direct evidence of a role for sphingomyelins in sunitinib-induced cardiotoxicity. This is accordant with previous reports of a decrease in sphingomyelin in isolated primary rat cardiomyocytes exposed to doxorubicin, a model cardiotoxin, concurrently with an increase in ceramide levels[59]. Furthermore, sphingomyelins, through their catabolism to ceramides, have been linked to the induction of apoptosis[58,60]. Consistent with sunitinib-dependent sphingomyelin perturbation at a site of toxicity, significant increases of some sphingomyelins were measured in the plasma of exposed rats on day 15. Assuming circulating levels of sphingomyelins are a direct consequence of cardiac changes, these discoveries suggest sphingomyelins could be successfully employed as biomarkers for drug-induced cardiotoxicity. Supporting this, previous studies have reported increases in circulating levels of sphingomyelins following myocardial infarction, and demonstrated a direct correlation with clinical biomarkers of cardiac pathologies, including high-sensitive-troponin and C-reactive protein, and lower left ventricular ejection fraction (LVEF)[61,62].

We also discovered a direct association of KU60648 levels and acylcarnitines. Since acylcarnitines have previously been proposed as biomarkers of dysregulated fatty acid metabolism and/or mitochondrial dysfunction[63,64], a commonly proposed mechanism in the progression of xenobiotic-induced cardiotoxicity[65], our discovery implicates mitochondrial dysfunction in the progression also of KU60648-induced cardiotoxicity. Consistent with this direct association between internal dose and response at the site of toxicity, large and significant increases of acylcarnitines were measured in the plasma of exposed rats on day 2 and 4. Significant increases in levels of acylcarnitines have been associated with heart failure, including correlation with LVEF[66], while it has also been reported that plasma acylcarnitine levels reflect the acylcarnitine profile in the cardiac tissues[67]. Taken together, these discoveries and previous findings identify acylcarnitines as tantalising putative biomarkers for drug-induced cardiac metabolic dysfunction. Further investigations would be required to test these hypotheses.

Following untargeted UHPLC-MS analysis of human plasma samples, we also present the capability of the untargeted ADME/TK workflow to reveal exposure to (suspect) pharmaceuticals. Specifically, we confidently detected seven of the UK NHS top 20 most prescribed pharmaceuticals within the dataset. Through subsequent implementation of the workflow, we detected and annotated 47 biotransformation products of the parent substances. Thus, our approach revealed insights into the pharmaceutical exposomes of the tested individuals. There is opportunity for such measurements (i.e., relative internal dose of pharmaceuticals and their biotransformation products) to be associated with (relative) levels of endogenous metabolites and lipids, measured in the same analysis, as an approach to relate exposure and internal fate to biological effect.

There are some limitations to the workflow presented here that require discussion. First, the workflow uses datasets acquired by UHPLC-MS-based metabolomics, i.e., analytical methods which are optimised to measure biochemicals using electrospray ionisation (ESI). However, many xenobiotics have physicochemical properties that are not compatible with ESI and will not be detectable[4]. As biotransformation products are typically more polar than parent substances, the detectability of such products using UHPLC-MS based metabolomics is anticipated to be higher. Furthermore, xenobiotics and their biotransformation products can occur at very low concentrations in biological samples, particularly when originating from unintended environmental exposure[11]. Detection of low abundant compounds is challenging, particularly in complex matrices from crude extraction of biological samples[4]. Where biotransformation products are measured, only partial structural formulae may be assigned. Finally, the workflow is limited to discovering only features related to either pre-defined exposure xenobiotics or suspects with available analytical standards.

In conclusion, we first demonstrated the capability to discover information on the disposition of pharmaceuticals from untargeted metabolomics using two case studies: sunitinib and KU60648 exposure in rats. Use of xenobiotic-related measurements revealed by our workflow has supported previous findings, providing confidence in our approach, and made discoveries on the biotransformation, tissue distribution and temporal changes in relative systemic exposure of the pharmaceuticals in exposed rats. Further application of the workflow on an untargeted metabolomics experiment with hiPSC-CMs has revealed the ability to discover metabolic competencies of in vitro model systems. We have shown that internal dose measurements made by untargeted metabolomics can aid the discovery of endogenous biochemical responses that are directly associated with internal exposure. Finally, we present how our approach can reveal insights into the exposome of humans, including mapping the biotransformation pathways of detected pharmaceuticals.

## Methods

### Chemicals

Sunitinib malate was purchased from Carbosynth Ltd (UK). KU60648 was synthesised in-house at AstraZeneca. Amitriptyline hydrochloride, lansoprazole, Paracetamol (acetaminophen) and ramipril were purchased from Merck Life Sciences UK Ltd. The purity of all chemicals was ≥98%.

### In vivo rat studies: animals, treatment and sampling

Animal studies were performed in accordance with the United Kingdom Animal (Scientific Procedures) Act 1986, were subject to local ethics committee (AstraZeneca Animal Welfare Review Board and Babraham Institute Animal Welfare and Ethical Review Board) approval and in line with project and personal license conditions.

Rats (Han-Wistar−Crl:WIST; 240–260 g or 220 − 240 g for male and female, respectively; age ca. 7 weeks) were purchased from Charles River Laboratories (CRL) UK. Animals were housed as previously described[68].

Animals were grouped randomly for the purposes of the study: groups 1–4 were formed of $N = 5$ male rats, groups 5–8 were formed of $N = 5$ female rats. Sunitinib and KU60648 were formulated in 20% w/v aqueous (2-hydroxypropyl)-β-cyclodextrin (HP-β-CD) and administered orally. Sunitinib-exposed rats (groups 6 and 8) were dosed with 25 mg/kg/day (10 mL/kg/day) of sunitinib malate for 14 days. KU60648-exposed rats (groups 2 and 4) were dosed with 150 mg/kg/day (10 mL/kg/day) for 2 days, then with 225 mg/kg/day (10 mL/kg/day) on day 3. These doses were selected as doses that were likely to elicit toxicologically-relevant molecular responses whilst considered suitable for investigation based on previous studies demonstrating tolerance to these doses, i.e., approximately the maximum tolerated dose.

Toxicokinetic analysis was carried out on plasma samples collected on day 1 (groups 2, 4, 6 and 8) at 1, 4, 6 and 24 h post-dose, day 3 (groups 2 and 4) pre-dose and 4 h post-dose and day 14 (groups 6 and 8) at 1, 4, 6 and 24 h post-dose. Sample collection and analysis was performed as described previously[68].

Blood samples for untargeted metabolomics were taken on days 1 and 2 (groups 1–4), day 1 and 4 (groups 5–6) or days 2 and 8 (groups 7–8) at 4 h post-dose. Terminal samples were also taken 24 h post-dose (day 4, groups 1–4) and 28 h post-dose (day 15, groups 5 − 8). 300 μL whole blood was collected via the tail vein, mixed with Lith Hep anticoagulant, and used to derive -150 μL of plasma for analysis.

KU60648-exposed rats (groups 2 and 4) and corresponding vehicle controls (groups 1 and 3) were terminated on day 4. Sunitinib-exposed rats (groups 6 and 8) and corresponding vehicle-controls (groups 5 and 7) were terminated on day 15. Following termination, the whole heart was removed (groups 3, 4, 7 and 8), with the apex sectioned for untargeted metabolomics analysis.

### Cardiomyocyte study

Human induced pluripotent stem cell-derived cardiomyocytes (hiPSC-CMs, Cellular Dynamics International, Fujifilm, USA) were defrosted according to manufacturer's instructions and plated at 500,000 cells per well in six-well plates pre-coated with 0.1% gelatin. (Each well formed a time-point, $N$, and was repeated from three separate vials). After plating in Cardiomyocyte plating media, cells were cultured in Cardiomyocyte maintenance media from 48 h post plating, until end of experiment. Ten days post-plating, the cardiomyocytes were treated with either 0.1% DMSO (control) or 5 μM sunitinib (treated), for either 1, 6 or 24 h, added as part of a media change. This concentration was selected as one that would elicit a toxicologically relevant molecular response without significant loss in cell viability, i.e., approximately the maximum tolerated dose, based on previous studies. Once the exposure time had elapsed, plates were placed on wet ice, the media was removed (to microcentrifuge tubes), then wells were washed twice with 1 mL cold 0.9% NaCl. Plates were then immediately frozen on dry ice. Media samples were spun at 10,000-$g$ for 5 min at 4 °C to remove any cell debris. The

supernatants were then transferred to clean microcentrifuge tubes. All samples were stored at −80 °C prior to metabolomic analyses.

### Human plasma samples

Plasma from 21 healthy human volunteers (gender, self-reported: 18 female, 3 male; age: 24–62) was obtained from the University of Birmingham's Human Biomaterials Resource Centre (HBRC), which holds ethical approval from an NHS Research Ethics Committee (NRES Committee North West – Haydock; Ref 20/NW/0001) to provide human biomaterials and associated data for a broad spectrum of biomedical research. Human biomaterials and associated data were obtained in accordance with the Human Tissue Act 2004 and associated Codes of Practice, and project specific use of human biomaterials and associated data were subject to the HBRC Access Review panel for ethical approval and sponsorship under the UK Policy Framework for Health and Social Care Research. All donors gave informed consent. They were not compensated.

Plasma samples selected for analysis were from donors (healthy volunteers) who reported to have taken some common medication in the 24 h prior to sampling. Plasma was prepared from peripheral blood samples mixed with EDTA and stored at −80 °C until analysis.

### Extraction of polar metabolites and lipids from cardiac tissue

Sample preparation was carried out according to previous studies[69,70], with some minor changes. Tissue sizes ranged from 54–124 mg. 8 μL/mg wet tissue mass of ice-cold methanol (LC-MS grade, VWR International, UK) and 3.2 μL/mg ice-cold water (LC-MS grade, VWR International, UK) was added to frozen tissue samples, which were homogenised in a bead-based homogenisation system (Precellys 24 with CK28 tubes, Stretton Scientific, UK). The homogenate was transferred to a 1.8 mL glass vial and 8 μL/mg ice-cold chloroform (HPLC grade, Fisher Scientific, UK) and 4 μL/mg water were added. Sample was vortexed (30 s), left on ice (10 min, for extraction) and centrifuged (2500-$g$, 4 °C, 10 min). Sample was set at room temperature (-20 °C) for 5 min to complete phase partitioning. Fixed volumes of the polar (400 μL−equivalent to 30 mg extracted tissue) and non-polar (250 μL−equivalent to 30 mg extracted tissue) were removed and dried in a SpeedVac sample concentrator (Savant SPD111V230, Thermo Fisher Scientific) or a nitrogen blow down drier (Techne FSC400D, Thermo Fisher Scientific), respectively. Samples were stored at −80 °C until analysis. To create polar intra-study quality control samples (QCs) an extra 300 μL of the polar extract (post-bi-phase partition) was taken from each sample, mixed (vortexed 30 s) and aliquoted (400 μL) before drying by SpeedVac. To create non-polar QCs an extra 200 μL of the non-polar extract (post-bi-phase partition) was taken from each sample, mixed (vortexed 30 s) and aliquoted (250 μL) before drying by nitrogen blow down drier. Extract blank samples were created by carrying out the above procedure in the absence of tissue. Prior to UHPLC-MS analysis, dried samples were resuspended in 300 μL 3:1 acetonitrile:water (polar extracts) or 300 μL 3:1 isopropanol:water (non-polar extracts), vortexed (30 s), centrifuged (20,000-$g$, 4 °C, 20 min) and 100 μL supernatant loaded into a low recovery HPLC vial (Chromatography Direct, UK).

### Extraction of polar metabolites and lipids from cardiomyocytes

Six-well plates (containing saline-washed cardiomyocytes with all liquid removed) were placed on dry-ice and 600 μL of 2:0.8 methanol:water (prechilled on dry ice for 60 min) was added to each well. Cardiomyocytes were dislodged into the solvent using a cell scraper (Corning) and all cardiomyocytes and liquid were removed into a fresh 1.8 mL glass vial. A further 240 μL of prechilled 2:0.8 methanol:water was added to the well, scraped and all contents added to the 1.8 mL glass vial. 600 μL ice-cold chloroform and 300 μL ice-cold water were added to the 1.8 mL glass vial and sample was vortexed (30 s), left on ice (10 min, for extraction) and centrifuged (2500-$g$, 4 °C, 10 min). Sample was set at room temperature (-20 °C) for 5 min to complete phase

partitioning. Fixed volumes of the polar (900 μL) and non-polar (600 μL) were removed and dried in a SpeedVac sample concentrator (Savant SPD111V230, Thermo Fisher Scientific) or a nitrogen blow down drier (Techne FSC400D, Thermo Fisher Scientific), respectively. Samples were stored at −80 °C until analysis. Extract blank samples were created by carrying out the above procedure in the absence of cells. Prior to UHPLC-MS analysis, dried samples were resuspended in 150 μL 3:1 acetonitrile:water (polar extracts) or 150 μL 3:1 isopropanol:water (non-polar extracts), vortexed (30 s), centrifuged (20,000-*g*, 4 °C, 20 min) and 100 μL supernatant loaded into a low recovery HPLC vial (Chromatography Direct, UK). To create the QCs, the remaining 50 μL liquid from each centrifuged resuspended sample was pooled (for polar and non-polar separately), vortexed and moved to a low recovery HPLC vial (Chromatography Direct, UK) for direct analysis.

### Extraction of polar metabolites and lipids from biofluids

Samples were prepared in accordance with previous studies[54], with some minor changes. Biofluids (rat plasma, human plasma or cardiomyocyte spent culture media) were thawed on ice and briefly vortexed (5 s). 50 μL of the biofluid was mixed with either (i) 150 μL 100% ice-cold acetonitrile (LC-MS grade, VWR International; polar metabolite extraction for HILIC analysis], or (ii) 150 μL 100% ice-cold isopropanol (LC-MS grade, VWR International; lipid extraction for RP C$_{18}$). Samples were vortexed and centrifuged (20,000-*g*, 4 °C, 20 min) and 100 μL supernatant removed into a low recovery HPLC vial (Chromatography Direct, UK) for direct analysis. To create the QCs, 50 μL of each plasma sample (plasma QC) or 50 μL of each media sample (media QC) was pooled, vortexed (30 s) and split into several 50 μL aliquots. Each aliquot was prepared as for the samples (above).

### Untargeted ultra-performance liquid chromatography-mass spectrometry

Samples were analysed as described previously[54] using a Q Exactive Focus Orbitrap MS (Thermo Scientific, Hemel Hempstead, UK) coupled to a Dionex Ultimate 3000 UPLC (Thermo Scientific), employing HILIC and RP C$_{18}$ chromatography. Instruments were controlled using XCalibur software (Thermo Scientific). For the HILIC method, an Accucore 150 Amide column (100 × 2.1 mm, 2.6 μm, Thermo Scientific) was used. Mobile phase A was 95% acetonitrile/water (10 mM ammonium formate, 0.1% formic acid) and mobile phase B was 50% acetonitrile/water (10 mM ammonium formate, 0.1% formic acid). The gradient was as follows: *t* = 0.0, 1% B; *t* = 1.0, 1% B; *t* = 3.0, 15% B; *t* = 6.0, 50% B; *t* = 9.0, 95% B; *t* = 10.0, 95% B; *t* = 10.5, 1% B; *t* = 14.0, 1% B. All changes were linear (curve = 5). The flow rate was 0.50 mL/min and the column temperature 35 °C. For the RP C$_{18}$ chromatography, a Hypersil GOLD C$_{18}$ column (10 × 2.1 mm, 1.9 μm, Thermo Scientific) was employed. Mobile phase A was 60% acetonitrile/40% water (10 mM ammonium formate, 0.1% formic acid) and mobile phase B was 85.5% propan-2-ol/9.5% acetonitrile/5% water (10 mM ammonium formate, 0.1% formic acid). The gradient was as follows: *t* = 0.0, 20% B; *t* = 0.5, 20% B, *t* = 8.5, 100% B; *t* = 9.5, 100% B; *t* = 11.5, 20% B; *t* = 14.0, 20% B. All changes were linear (curve = 5). The flow rate was 0.40 mL/min and the column temperature 55 °C. In all cases analysis was performed in positive and negative ionisation modes separately at a resolution of 70,000, between 70 and 1050 *m/z* (HILIC) and 150–2000 *m/z* (RP C$_{18}$). The sample injection volume was 2 μL. MS$^2$ fragmentation data was collected by data-dependent acquisition (DDA using 'Discovery Mode') of QCs using HCD with stepped collision energies (CEs) (HILIC 25, 60, 100; RP C$_{18}$ 20, 50, 80). For HILIC and RP C$_{18}$ separately, MS$^2$ data were collected for three different *m/z* ranges from three separate injections. Scan ranges were: HILIC *m/z* 70–200, *m/z* 200–400, and *m/z* 400–1000; C$_{18}$ RP *m/z* 200–400, *m/z* 400–700 and *m/z* 700–1500. Additional MS$^2$ fragmentation data of putative xenobiotic-related features was collected by DDA, using targeted inclusion lists across multiple injections of selected xenobiotic-exposed samples, using

HCD with stepped CEs (HILIC 25, 60, 100; RP C$_{18}$ 20, 40, 100). For MS$^2$ fragmentation data acquisition, analysis was performed at a of resolution of 35,000 and 17,500 for full scan (MS$^1$) and MS$^2$, respectively.

The analysis of human plasma extracts was performed using a positive ion HILIC UHPLC-MS method only (based on results of rat studies above) as described above, but deployed on an Orbitrap ID-X Tribrid MS (Thermo Scientific) coupled to a Vanquish Horizon UHPLC (Thermo Scientific) (with method gradient: *t* = 0.0, 1% B; *t* = 2.1, 1% B; *t* = 4.1, 15% B; *t* = 7.1, 50% B; *t* = 10.1, 95% B; *t* = 11.0, 95% B; *t* = 11.5, 1% B; *t* = 15.0, 1% B). Analysis was performed at a resolution of 120,000, over a scan range of m/z 70–1050. MS$^2$ fragmentation data was collected by DDA (ddMSnScan) from pooled samples using HCD with stepped NCEs (20, 40, 130%), and inclusion lists targeting SyGMa predicted biotransformation products ([M + H]$^+$, [M + Na]$^+$ and [M + NH$_4$]$^+$ ion forms). Analysis was performed at resolutions of 60,000 and 30,000 for full scan and MS$^2$, respectively, for fragmentation data acquisition.

### Untargeted metabolomics raw data processing

Vendor format raw data files (.RAW) were converted to mzML file format using ProteoWizard software[71]. Full scan (MS$^1$) data deconvolution was performed by XCMS operated in Galaxy[51], as reported previously[54] (settings: min. peak width (HILIC = 4; RP C$_{18}$ = 6); max. peak width (30); ppm (HILIC = 12; RP C$_{18}$ = 14); mzdiff (0.001); bw (0.25); mzwid (0.01); minfrac (0.2 for rat and cardiomyocyte datasets, 0.05 for human dataset)). A data matrix of peak intensities for metabolite features (*m/z*-retention time pairs) vs. samples were constructed. MS$^2$ data files were processed, filtered and averaged using the R/Bioconductor package msPurity[53] (settings: XCMS (as described above except minfrac (0.1)), plim (0.5), ppm (5.0)). Processed MS$^2$ spectra were aligned to metabolite features in the peak matrix generated from full scan (MS$^1$) data files using 5 ppm mass error and 10 s retention time tolerance window.

### Xenobiotic-based data analysis and annotation

Exposure to suspect xenobiotics, i.e., any of the top 20 pharmaceuticals prescribed by the UK NHS between 2014 and 2020, was defined for human samples with unknown exposures as follows. First, measured features corresponding to each of the ten suspect xenobiotics (pharmaceuticals) were identified by (i) match of measured *m/z* to theoretical *m/z*, for a low confidence annotation (MSI level 3/Schymanski level 3), (ii) match of selected MS$^2$ fragmentation data to MS$^2$ spectrum contained within a public database (MassBank, GNPS), for a mid-confidence annotation (MSI level 2/Schymanski level 2) and/or (iii), match of MS$^2$ fragmentation pattern and retention time to that of a chemical standard measured using the same instrument and analytical method, for a high-confidence annotation (MSI Level 1/Schymanski level 1). Using the annotated features, samples were defined as exposed to a given xenobiotic if the relative intensity of the representative feature was >10-fold the median intensity across all samples and QCs, after imputing missing values with the lowest measured relative intensity across all features. Where relative intensity was below this value, the sample was defined as 'control'.

In the case of laboratory model-based experiments (in vivo rat and in vitro cardiomyocyte studies), the 'exposed' and 'control' sample labels were defined as per the experimental conditions.

Putative xenobiotic-related features were discovered by applying the following filters: feature present in ≥80% (for rat and cardiomyocyte datasets) or ≥50% (human datasets) of exposed samples; feature present in ≤50% biological control samples; and the median intensity of a feature in exposed samples is ≥10-fold its median intensity in biological control samples. This was implemented using the R/Bioconductor package structToolbox[52] (v1.6.0, https://bioconductor.org/packages/release/bioc/html/structToolbox.html). Ion form annotation and feature grouping of the putative xenobiotic-related features was carried out using the functions 'Group features' and 'Annotate Peak

Patterns' in the python package BEAMSpy[50] (v1.1.0, https://github.com/computational-metabolomics/beamspy), using 5 ppm mass error, 5 s retention time tolerance window.

Pearson's correlation analysis was carried out using the R/cran package RcmdrMisc (https://cran.r-project.org/web/packages/RcmdrMisc/index.html) on combined data from all UHPLC-MS assays following normalisation of intensity measurements by probabilistic quotient normalisation (PQN), applying coefficients calculated for the endogenous datasets. A given pair of features were defined as co-responsive when Holm adjusted $p$ value < 0.05 and $R \geq 0.75$. A network diagram of the correlation matrix was analysed and visualised in Cytoscape 3.7.2[72].

Biotransformation products of parent substances (sunitinib, KU60648, paracetamol, omeprazole, atorvastatin, lansoprazole, bisoprolol, and amitriptyline) were predicted using SyGMa[37] and the 'Generate Expected Compounds' tool in Compound Discoverer (v3.0, Thermo Scientific, suntinib and KU60648, only). For SyGMa, both number of Phase I and Phase II transformation steps were set to 3. The outputs were filtered to predictions with a SyGMa score ≥0.01% and predictions where molecular formulae of predicted biotransformation products were identical were combined. For Compound Discoverer predictions, the standard software transformation library was used to predict biotransformation products for both xenobiotics, with maximum number of dealkylation steps set to 1, maximum number of Phase II steps set to 1 and maximum number of all steps set to 3. Predicted biotransformation products were aligned to putative xenobiotic-related features using the R programming language (https://www.R-project.org), using 5 ppm mass error.

The molecular formula/structures associated with the measured fragmentation peaks of parent substances and their putative biotransformation products were annotated in silico using MetFrag in command line[40] with the candidate molecule(s) a given spectra user-defined as the annotation from SyGMa and/or Compound Discoverer, or parent substances where features were not annotated (settings: fragment peak match mass deviation = 5 ppm)

K-means cluster analysis, a type of unsupervised clustering, was carried out using the R programming language (https://www.R-project.org). Prior to analysis, data was scaled by unit variance. An elbow plot was generated for each dataset to select the optimal value of k prior to execution of k-means cluster analysis using the median intensity of feature across biological replicates at each time point.

### Endogenous metabolite and lipid-based data analysis
Prior to data analysis, datasets were filtered as follows: any feature whose median intensity in biological samples is <20× its median intensity in blank samples was removed; features with relative standard deviation ≥ 30% across the QCs were removed; samples with >50% missing values were removed; features which were missing in ≥10% QCs and/or ≥50% of all samples were removed; features present in the list of putative xenobiotic features were removed. Univariate statistical analysis (t-test) were applied after PQN. Principal components analysis was carried out after PQN, missing value imputation by k-nearest neighbour ($k = 5$) and generalised log transformation. These steps were executed using the R/Bioconductor packages pmp (v1.6.0, https://bioconductor.org/packages/release/bioc/html/pmp.html) and structToolbox[52] (v1.6.1, https://bioconductor.org/packages/release/bioc/html/structToolbox.html).

Endogenous polar metabolite annotation was performed using Compound Discoverer (Thermo Scientific). Metabolite features within the UHPLC-MS$^2$ data were searched against the mzCloud database. Only annotations with spectral match score >0.6 were used. Endogenous lipid annotation was performed using LipidSearch (Thermo Scientific). Lipid features within the UHPLC-MS$^2$ data were searched against the entire in silico HCD database (5 ppm mass error). Only annotations graded A–C were used for annotation purposes (Grade A—all fatty acyl chains and class were completely identified; Grade B—some fatty acyl chains and the class were identified; Grade C—either the lipid class or some fatty acyls were identified). These approaches provided annotations consistent with reporting level 2 of the MSI recommendations[43]. Annotations were aligned to the XCMS outputs using the R programming language (https://www.R-project.org), using 3 ppm mass error and 10 s retention time tolerance window.

Spearman's correlation analysis of annotated lipids and parent xenobiotics (sunitinib or KU60648) in the cardiac tissue of exposed rats was carried out using PQN-normalised peak intensity measurements. This was executed using the R/cran package RcmdrMisc (https://cran.r-project.org/web/packages/RcmdrMisc/index.html). A network diagram of the resulting correlation matrix was analysed and visualised in Cytoscape 3.7.2[72]. Over-representation analysis of lipid classes significantly correlated with sunitinib or KU60648 ($p < 0.05$) was carried out by one-way Fisher's Exact test, using full list of annotated lipids as the reference set.

### Reporting summary
Further information on research design is available in the Nature Portfolio Reporting Summary linked to this article.

## Data availability
Untargeted UHPLC-MS(/MS) metabolomics raw and derived data and associated metadata that support the findings of this study have been deposited in MetaboLights with the accession code MTBLS2746. Additionally, fragmentation data (MS/MS) used to support findings presented in the manuscript has been deposited in MassBank (https://massbank.eu/MassBank/; accession codes: MSBNK-UoB-XB000xxx, where xxx is 101–112, 200–215, 300–306, 400–406, 500–504, 600, 700–701, 800 or 900–902). Source data are provided with this paper.

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

## Acknowledgements

We gratefully acknowledge the contribution to this publication made by the University of Birmingham's Human Biomaterials Resource Centre which has been supported through Birmingham Science City—Experimental Medicine Network of Excellence project. The authors thank Thermo Fisher Scientific for their support of this project via the University of Birmingham—Thermo Fisher Scientific Technology Alliance Partnership. We also thank Dr. Lukáš Najedkr and Dr. Andris Jankevics (Phenome Centre Birmingham, UK) for their scientific advice. We thank the BBSRC (BB/S507064/1) and AstraZeneca for funding this project via a PhD studentship awarded to T.J.B.

## Author contributions

T.J.B. developed the workflow, analysed the data and prepared the manuscript. A.D.S. performed the laboratory experiments and provided conceptual advice. A.R.H. performed the laboratory experiments. R.J.M.W. provided technical support and conceptual advice. G.R.L. provided technical support and statistical advice. R.M. co-designed and directed the in vivo studies. A.W. co-designed the experiments and gave conceptual advice. A.P. co-designed the experiments, gave conceptual advice and co-supervised the project. M.R.V. developed the concept, co-designed the experiments and supervised the project. All authors edited and approved the manuscript.

## Competing interests

The authors declare no competing interests.
