## [Peer Review File · Nature Communications]

REVIEWER COMMENTS

Reviewer #1 (Remarks to the Author):

Thank you for giving me the opportunity to review this interesting work. Bowen et al. used a well-established analytical untargeted pipeline for investigating the biotransformation and kinetics of two drugs in rat plasma, cardiac tissue and cell-based samples. The paper is well-structured and written (great job; it seems to be the first paper of the first author). The lab experiments are technically sound and mostly reported in sufficient detail to understand what was done (as can be expected from a renown and respected group such as the one of Dr. Viant).

There is great value in applying untargeted, high-resolution mass spectrometry-based metabolomics to discover xenobiotic biotransformation and directly link exposure to a biological response/effect. It is well-known that untargeted metabolomics can be useful in annotating and identifying biotransformation products of xenobiotics including drugs, environmental toxicants and food-related molecules. Also the combined exposure/response readout of such experimental design has been proposed before, often in the context of exposome research or systems toxicology.

The main issue is that this paper (including the title) suggests a generic metabolomics platform for xenobiotics in toxicological research. However, in its current form the manuscript is purely related to drugs and pharmacokinetics rather than to toxins/toxicants (and toxicokinetics) that we are exposed to unintentionally through the environment, resulting in much lower exposure. Therefore, this approach seems limited to high exposure levels in lab models. It might be much broader and more useful if the authors could demonstrate convincingly that it can be applied to environmental toxicants at realistic concentrations in human in vivo samples. This referee is not fully convinced by extracting biotransformation data only from animal/cell models that have been dosed (i) with drugs and (ii) at extremely high concentrations. Sure, the putative structures elucidated in Figures 2 and 3 are very comprehensive (nice job) but this is what one would expect when using high doses, a state-of-the-art HRMS instrument and four separate analytical runs.

The authors highlight that they implemented a novel workflow with regards to xenobiotics-related signals in the MS data. This reviewer believes that there is value in the proposed approach but that the level of technical novelty is rather limited and most parts of the general workflow are kind of standard in untargeted metabolomics or rely on tools/algorithms that have been used before. Nevertheless, they have certainly been combined in a useful way.

Finally, I would like to mention that the underlying raw, meta, and QC data was not fully provided via a public repository as it should be done according to FAIR principles. Therefore, the reviewer was unable to judge its quality.

Specific comments:

- Introduction: Concise and with some highly relevant statements (e.g. line 47-50). Generally, I would suggest to cut the number of references a bit and focus on those with special relevance (e.g. line 51 and 57). Also, some current papers that take advantage of ^{13}C -labeling for investigating unknown biotransformation products that are relevant in this context are missing.

- ,... to measure thousands of low molecular weight biochemicals (metabolites and lipids) in a biological sample'. We can measure thousands of features but in most cases far less metabolites. I suggest to tone down a bit as statements like this frequently result in unmet expectations by non-experts what untargeted metabolomics can realistically deliver.

- Page 9: As mentioned above, I think the paper would clearly benefit if you could test the workflow in human in vivo samples. Since some of the co-authors are from a pharmaceutical company I wonder if plasma from clinical trials is available? Would be far more interesting than rat plasma; biotransformation in rodents is often completely different than in humans. I would further suggest to expand the approach to environmental/occupational exposures to see if you can identify so far unknown biotransformation products in human plasma and/or urine. This would be a true breakthrough and more be in line with what I expected after reading the title

- Figure 2: Please include the information given in c (ESI mode and chromatography) and d (phase I vs. II) also in f. This would allow to judge the feasibility in a more complete way (e.g. I would have expected that the glucuronide and sulfate conjugates can be seen in both, HILIC negative and RP negative)

- I am wondering why no reference standard have been included to yield level 1 identifications (besides the native drug)? A quick web search yielded a couple of vendors offering at least some important ones. I would also expect that the industrial collaborator might have standards that would further boost the confidence in the results and allow for absolute quantitation

- On page 12 the authors talk about ^{14}C -radiolabeling as the gold standard in pharmacokinetics. This is of course true; the paper may benefit from a short discussion that this approach is not needed anymore in many cases with the new tools using ^{13}C and deuterium labels developed in recent years. Moreover, HRMS is highly sensitive allowing much lower concentrations (that are more realistic in humans) to be used. Just an idea in line with the workflow proposed by the paper.

- Figure 3: Very similar to Figure 2. I would like to see some peaks and spectra of low abundance metabolites

- Figure 4: The error bars in b seem a bit odd at first glance. Please also explain how the clustering in c was done (rational for clustering certain metabolites together)

- The ,endogenous metabolomic response to xenobiotic exposure' part seems a bit underdeveloped. Changes in lipids are shown and discussed but I assume that much more should be going on with regards to other endogenous small molecules?
- Figure 6: Again, it would be very good to see here data on human plasma
- I really like the discussion in line 530-536; very relevant
- In the conclusions (line 557) I miss the limitations of the workflow
- Methods:
 - Please comment on the purity of the two drugs used. We frequently see that these standards are contaminated with potential metabolites at very low levels
 - Dosing: 25-225 mg/kg/d seem very high. Important to highlight that environmental toxicants are mostly present in the human body at lower concentrations by a factor of maybe 100k – 1M x
 - Rat experiments: Did you use metabolic cages and are corresponding urine and stool samples available? Would be interesting to screen for biotransformation products there as well; especially for sunitinib that is mainly excreted via feces
 - Cell model: Also here the concentration is high (5 μ M); I suggest to redo the experiment at a 1,000 and 10,000 x lower dose and compare the results to challenge the workflow
 - I guess the mobile phase A as 60% acetonitrile and 40% water and not 60% acetonitrile/water?
 - It would be great if you could upload all MS2 spectra of the newly discovered biotransformation products to open spectral libraries such as GNPS and MassBank. This would enable fellow researchers to capitalize on the new structures that have been annotated
 - I am wondering how much effort it would be for any ,standard' metabolomics lab to implement this workflow? A short statement would be appreciated.

Reviewer #2 (Remarks to the Author):

Authors present an untargeted metabolomics workflow that simultaneously considers exogenous metabolites together with endogenous metabolic responses. This type of systematic and agnostic approach is critical to better understanding the diversity of chemicals to which we are exposed and the associated biological responses. Recognition of this comprehensive assessment of environmental factors is central to the exposome. There have been significant methodological developments over the past ~15 years since the exposome concept was originally presented. It was surprising that the authors did not highlight existing work in this field (i.e. Vermeulen et al. 2020 Science. 367(6476):

392.) Review of the related literature would help refine the proposed workflow to be consistent with similar approaches that been implemented by others. Some specific comments follow:

- 1) In defining criteria for filtering data, authors specify that xenobiotic-related features should be present in all samples. This requirement assumes of a case and control, limiting application of the workflow to human samples without defined exposures.
- 2) Suggestion to consider the published literature when evaluating the confidence of identified xenobiotics (i.e. Schymanski et al. 2014 EST. 48(4): 2097.)
- 3) Two case studies are presented. Each case evaluates rats exposed to a single drug with biotransformed products presented. To demonstrate how this type of exposomic workflow could be leveraged, it would be useful to apply the method to samples with unspecified exposure mixtures.

Reviewer #3 (Remarks to the Author):

This fascinating manuscript describes a workflow for “untargeted toxicokinetics” using ultra-high performance liquid chromatography mass spectrometry (UHPLC-MS). As opposed to traditional toxicokinetics in which the chemical analysis focuses on a parent compound and, perhaps, a few key metabolites, this method allows identification and tracking of a xenobiotic signature of multiple metabolites. The authors discuss how this method would allow for metabolite discovery, inform evaluation of metabolism in vitro, and enhance analysis of the impact of chemicals on endogenous biochemistry.

I found the manuscript to be both scientific valuable and generally very interesting. This is a complex analysis and I hope that most of my comments are taken as suggestions for enhancing the ease of understanding by a broader audience but not as negative comments. First, the various sections read a bit like loosely connected vignettes rather than as a single narrative. I think this is because there are many implications to the method and, to the authors credit, they have explored several.

My major comments follow:

To help with understanding everything that was done, I suggest adding two tables. First, a table describing the two primary chemicals studied might be helpful: potential columns might include # of

predicted metabolites, # of analytes probable with targeted methods (that is, with available standards), # of analytes detected with workflow presented, estimated half-life in relevant species. Second, and more importantly, a table describing the datasets analyzed would be very helpful. I would appreciate known the reference, chemical, in vitro/in vivo, species, doses, measured time points, and the reason for the analysis (such as “Development of signature”, “Evaluation of time course methods”, “Evaluation of metabolic in vitro competency”) all in one place.

One more extreme approach to simplify the paper would be removing the sections related to subtracting the xenobiotic “fingerprints” to enhance analysis of effects on endogenous chemicals. Intuitively this is an important application and I am glad that the authors discuss it, but I do not know that they actually demonstrate enhanced biological understanding here. Alternatively, the authors could provide contrast with analysis of the data without the xenobiotic signature removed to see if statistical insights are improved.

The following comments are minor:

I think the word “extract” (as in 168-169) may be confusing to some readers, especially since one of the analyses is removing the xenobiotic signature from the data. I suggest that the authors find an alternative (such as “identify features” or “isolate for analysis”). Perhaps with slight rephrasing they might use the term “feature extraction”.

On lines 299-300 I think the authors should be more explicit that they are contrasting conventional quantification (that is targeted analysis of 7(?) chemicals) in contrast to methods under study here.

On line 330 – “days 1 and 4” should be “days 1 and 14”

On 342 how many biotransformation products are tracked? The 22 from Figure 2f?

For Line 374 How were the “putative xenobiotic-related features” identified? I presume this is the signature from the in vivo data. However, since you discussing how much of that signature was observed in vitro (section Contrasting the metabolic competencies...) having a list of the total numbers would be helpful. This could be addressed in the second table I suggested.

Point-by-point response to the referees' comments.

Reviewer 1:

Thank you for giving me the opportunity to review this interesting work. Bowen et al. used a well-established analytical untargeted pipeline for investigating the biotransformation and kinetics of two drugs in rat plasma, cardiac tissue and cell-based samples. The paper is well-structured and written (great job; it seems to be the first paper of the first author). The lab experiments are technically sound and mostly reported in sufficient detail to understand what was done (as can be expected from a renown and respected group such as the one of Dr. Viant).

There is great value in applying untargeted, high-resolution mass spectrometry-based metabolomics to discover xenobiotic biotransformation and directly link exposure to a biological response/effect. It is well-known that untargeted metabolomics can be useful in annotating and identifying biotransformation products of xenobiotics including drugs, environmental toxicants and food-related molecules. Also the combined exposure/response readout of such experimental design has been proposed before, often in the context of exposome research or systems toxicology.

The main issue is that this paper (including the title) suggests a generic metabolomics platform for xenobiotics in toxicological research. However, in its current form the manuscript is purely related to drugs and pharmacokinetics rather than to toxins/toxicants (and toxicokinetics) that we are exposed to unintentionally through the environment, resulting in much lower exposure. Therefore, this approach seems limited to high exposure levels in lab models. It might be much broader and more useful if the authors could demonstrate convincingly that it can be applied to environmental toxicants at realistic concentrations in human *in vivo* samples. This referee is not fully convinced by extracting biotransformation data only from animal/cell models that have been dosed (i) with drugs and (ii) at extremely high concentrations. Sure, the putative structures elucidated in Figures 2 and 3 are very comprehensive (nice job) but this is what one would expect when using high doses, a state-of-the-art HRMS instrument and four separate analytical runs.

- The reviewer is correct that our initial deliberate focus was on intentional xenobiotic exposures. Indeed, they should note that neither of the words “environment” or “exposom*” occurred even once in our original manuscript. However, as requested, we’ve expanded our demonstration of how the workflow can reveal exposure to xenobiotics, mainly commonly prescribed drugs that have emerged as chemicals of concern (Meijer *et al.* 2021 - <https://doi.org/10.1016/j.envint.2021.106511>), plus their biotransformation products. The parent xenobiotics detected included Paracetamol, omeprazole, atorvastatin, lansoprazole, bisoprolol, and amitriptyline. We discovered a total of 46 biotransformation products of these xenobiotics using the untargeted ADME/TK workflow. This was achieved at ‘real-world’ concentrations in human *in vivo* plasma samples, as requested, following UHPLC-MS analysis. The new results demonstrate the workflow is not limited to “high exposure levels in lab models”. A

significant new results section (“Untargeted workflow discovers fate of xenobiotics in humans”) has been developed and added to the manuscript.

The authors highlight that they implemented a novel workflow with regards to xenobiotics-related signals in the MS data. This reviewer believes that there is value in the proposed approach but that the level of technical novelty is rather limited and most parts of the general workflow are kind of standard in untargeted metabolomics or rely on tools/algorithms that have been used before. Nevertheless, they have certainly been combined in a useful way.

- We agree that the individual elements of our proposed workflow are not novel, instead its the integration of a series of existing open-source softwares, tools and algorithms that has enabled the advances that we present in the manuscript. To ensure we do not mislead the reader, we have changed the wording in the abstract (and paper) from "novel workflow" to "assembled a workflow").

Finally, I would like to mention that the underlying raw, meta, and QC data was not fully provided via a public repository as it should be done according to FAIR principles. Therefore, the reviewer was unable to judge its quality.

- As requested, the substantial collection of 2442 raw and derived UHPLC-MS data files and their metadata that were used in this manuscript have been submitted to MetaboLights for curation (accession code: MTBLS2746). We will forward the link to access the repository to the editor as soon as curation by the MetaboLights team is complete and the repository can enter the “Review” phase.

Specific comments:

- Introduction: Concise and with some highly relevant statements (e.g. line 47-50). Generally, I would suggest to cut the number of references a bit and focus on those with special relevance (e.g. line 51 and 57). Also, some current papers that take advantage of ¹³C-labeling for investigating unknown biotransformation products that are relevant in this context are missing.

- Whilst the authors believe reference to regulatory guidance documents for the safety assessment of pharmaceuticals, biocides and industrial chemicals are required to reveal the importance of this work to multiple fields at the regulatory level, other references of lesser relevance have been removed.
- The authors recognise the importance of referencing recent developments in the field and have now cited studies implementing stable isotopic labelling for xenobiotic metabolite identification (lines 53 and 74).

- ,... to measure thousands of low molecular weight biochemicals (metabolites and lipids) in a biological sample’. We can measure thousands of features but in most cases far less metabolites. I suggest to tone down a bit as statements like this frequently result in unmet expectations by non-experts what untargeted metabolomics can realistically deliver.

- The authors appreciate this comment, though the actual number of polar metabolites detected is currently unknown, and large numbers of lipids are certainly readily

detectable. Hence toning down to a smaller defined number could be equally misleading. Given this uncertainty, we have changed the statement to “large numbers of low molecular weight biochemicals”

- Page 9: As mentioned above, I think the paper would clearly benefit if you could test the workflow in human in vivo samples. Since some of the co-authors are from a pharmaceutical company I wonder if plasma from clinical trials is available? Would be far more interesting than rat plasma; biotransformation in rodents is often completely different than in humans. I would further suggest to expand the approach to environmental/occupational exposures to see if you can identify so far unknown biotransformation products in human plasma and/or urine. This would be a true breakthrough and more be in line with what I expected after reading the title

- As described above, we've demonstrated the workflow can reveal exposure to xenobiotics and their biotransformation products at 'real-world' concentrations in vivo. Through application of our workflow, we discovered a total of 46 biotransformation products of 6 chemicals of emerging concern detected within the plasma of 'healthy' humans (new results section: "Untargeted workflow discovers fate of xenobiotics in humans").

- Figure 2: Please include the information given in c (ESI mode and chromatography) and d (phase I vs. II) also in f. This would allow to judge the feasibility in a more complete way (e.g. I would have expected that the glucuronide and sulfate conjugates can be seen in both, HILIC negative and RP negative)

- As requested, additional information has been added to the figure: the analytical method that detects the compound is displayed through use of symbols alongside the compound identity, and the type of transformation (i.e., phase I or phase II) is indicated by the colour of the arrow.

- I am wondering why no reference standard have been included to yield level 1 identifications (besides the native drug)? A quick web search yielded a couple of vendors offering at least some important ones. I would also expect that the industrial collaborator might have standards that would further boost the confidence in the results and allow for absolute quantitation

- The lack of use of reference standards specifically for the biotransformation products was an active decision by the authors. We seek to publish a paper that, while having broader applicability, is totally compatible with the application of safety testing of new chemical entities, including for many thousands of industrial chemicals. Reference standards of the biotransformation products of such new chemicals are very rarely available.

- On page 12 the authors talk about ¹⁴C-radiolabeling as the gold standard in pharmacokinetics. This is of course true; the paper may benefit from a short discussion that this approach is not needed anymore in many cases with the new tools using ¹³C and deuterium labels developed in recent years. Moreover, HRMS is highly sensitive allowing much lower concentrations (that are more realistic in humans) to be used. Just an idea in line with the workflow proposed by the paper.

- The authors appreciate this suggestion and have now presented both [¹⁴C]-labelling and stable isotope ([¹³C] / [²H]) labelling as methods to measure xenobiotic biotransformation

(lines 73-75). Additionally, the benefits that HRMS can provide, mainly enabling discovery of novel biotransformation products through high sensitivity, has been highlighted (lines 554-556 and 596-588).

- Figure 3: Very similar to Figure 2. I would like to see some peaks and spectra of low abundance metabolites

- Extracted ion chromatograms and MS² fragmentation spectra of parent xenobiotics and their biotransformation products (including those of lower abundance) from the analysis of rat plasma have been added to the supplementary materials (Supplementary Figures 2, 3, 4 and 5). Similarly, extracted ion chromatograms and MS/MS spectra of parent xenobiotics and their biotransformation products detected in human plasma (new results section) have also been added to the supplementary materials (Supplementary Figures 18 and 19).

- Figure 4: The error bars in b seem a bit odd at first glance. Please also explain how the clustering in c was done (rational for clustering certain metabolites together)

- It is not clear to us what the reviewer means by the error bars 'seem a bit odd'. The authors would be happy to edit should the reviewer wish to provide more detail on the issue.
- Clustering was performed by k-means clustering - a type of *unsupervised* cluster analysis; i.e. there is no rationale for clustering certain metabolites together. The type of clustering performed is detailed in the manuscript, both in the figure caption and in the Materials and Methods section (lines 915-919).

- The 'endogenous metabolomic response to xenobiotic exposure' part seems a bit underdeveloped. Changes in lipids are shown and discussed but I assume that much more should be going on with regards to other endogenous small molecules?

- The section titled "Lipid responses associate with internal xenobiotic levels." has been substantially developed in the revised manuscript. Mainly, our principal message of this section, that endogenous changes are best discovered following removal of the xenobiotic fingerprint, has been better demonstrated through comparison of principal components analysis scores and loadings before and after exclusion of the xenobiotic fingerprint (see new Figures 5a and b, Supplementary Figures 10 and 12).
- Furthermore, the endogenous response measured in the plasma is now presented, in addition to the cardiac endogenous response (see new Figure 5c, Supplementary Figure 15, Supplementary Tables 7, 9, 13, and 14).
- In the revised manuscript the consistencies across the separate analyses (including correlation analyses of cardiac tissue measurements, univariate statistics of cardiac tissue data, and univariate statistics of plasma data) are now highlighted and drawn upon, strengthening the identification of putative biomarkers for drug-induced cardiotoxicity.
- The response of the cardiac tissue (the site of toxicity) is dominated by lipid changes, hence the authors believe this is the most enticing finding and we focus on it (rather than "other endogenous small molecules") in the manuscript.

- Figure 6: Again, it would be very good to see here data on human plasma

- While we were unable to access or measure human plasma data for sunitinib, we have investigated several other xenobiotics in human plasma samples to demonstrate the applicability of the workflow to an additional sample type, specifically the ability to measure, extract and identify xenobiotics and their biotransformation products at real-world human in vivo concentrations.

- I really like the discussion in line 530-536; very relevant

- Thank you!

- In the conclusions (line 557) I miss the limitations of the workflow

- We have added some limitations of the workflow (lines 674-686), referring to chemical class compatibility with electrospray ionisation, the often very low abundance of xenobiotics and their biotransformation products and related sensitivity limitations, the challenge of complete structural annotations, and the reliance on availability of an analytical standard for the parent xenobiotic(s).

- Methods:

- Please comment on the purity of the two drugs used. We frequently see that these standards are contaminated with potential metabolites at very low levels

- Purity of all chemical standards was $\geq 98\%$. This information has been added to the manuscript (lines 703-704).

- Dosing: 25-225 mg/kg/d seem very high. Important to highlight that environmental toxicants are mostly present in the human body at lower concentrations by a factor of maybe 100k – 1M x

- The rat study was designed as a safety pharmacology biomarker discovery experiment. A dose that elicits a molecular response (e.g., molecular initiating event and molecular, subcellular and cellular level key events in development of toxicity) without an adverse event (e.g., based on clinical observations, food consumption, body weight loss), i.e., the maximum tolerated dose, was deemed appropriate to that end. This rationale for dose selection in the rat study has been added to the manuscript (lines 719-721).

- Rat experiments: Did you use metabolic cages and are corresponding urine and stool samples available? Would be interesting to screen for biotransformation products there as well; especially for sunitinib that is mainly excreted via feces

- Specifically, animals were “pair-housed in standard rat individually ventilated cages with a non-instrumented companion rat”. The methods (animal husbandry) were the same as reported in Adeyemi *et al.*, 2020 (<https://doi.org/10.1016/j.vascn.2020.106679>). This paper is cited in our manuscript (line 712).
- Urine and stool samples were not collected.

- Cell model: Also here the concentration is high (5 μM); I suggest to redo the experiment at a 1,000 and 10,000 x lower dose and compare the results to challenge the workflow

- The cardiomyocyte study was designed for the purpose of biomarker discovery, thus a high enough concentration to elicit a toxicological response without significant loss in cell viability, i.e., approximately the maximum tolerated concentration, was deemed appropriate. This rationale has been added to the manuscript (lines 744-746).
- Instead of redoing the *in vitro* experiment at artificially lower concentrations, we have instead focused on demonstrating the workflow using human *in vivo* samples to discover xenobiotics and their biotransformation products at actual 'real-world' concentrations.

- I guess the mobile phase A as 60% acetonitrile and 40% water and not 60% acetonitrile/water?

- That is correct. This has been clarified in the manuscript (line 829).

- It would be great if you could upload all MS2 spectra of the newly discovered biotransformation products to open spectral libraries such as GNPS and MassBank. This would enable fellow researchers to capitalize on the new structures that have been annotated

- MS/MS spectra corresponding to analytical standards of six of the parent xenobiotics (Sunitinib, KU60648, Amitriptyline, Lansoprazole, Paracetamol and Ramipril) have been uploaded to MassBank for public access (accessions: MSBNK-UOB-XB00000x, available to view here <https://github.com/MassBank/MassBank-data/pull/208>). These data will be merged with the MassBank-data main GitHub branch upon the author's request, which we intend to be simultaneous with publication of our manuscript.

- I am wondering how much effort it would be for any 'standard' metabolomics lab to implement this workflow? A short statement would be appreciated.

- We agree that would be helpful. A statement has been added to the discussion (lines 575-577).

Reviewer 2:

Authors present an untargeted metabolomics workflow that simultaneously considers exogenous metabolites together with endogenous metabolic responses. This type of systematic and agnostic approach is critical to better understanding the diversity of chemicals to which we are exposed and the associated biological responses. Recognition of this comprehensive assessment of environmental factors is central to the exposome. There have been significant methodological developments over the past ~15 years since the exposome concept was originally presented. It was surprising that the authors did not highlight existing work in this field (i.e. Vermeulen et al. 2020 Science. 367(6476): 392.) Review of the related literature would help refine the proposed workflow to be consistent with similar approaches that been implemented by others.

- The authors appreciate these comments, and although applicability of the workflow to exposomics was *not* initially intended, indeed the reviewer should note that neither of the words "environment" or "exposom*" occurred even once in our original manuscript, instead we focused on intentional chemical exposures, we agree that highlighting its applicability to exposomics is valuable. With our significant addition of a new section ("Untargeted workflow discovers fate of xenobiotics in humans") focused on human samples, this is now recognised in the manuscript, including revealing how application of our workflow can help characterise the exposome of individuals. Furthermore, existing

work in this field, including Vermeulen *et al.*, 2020, has now been appropriately referenced.

Some specific comments follow:

1) In defining criteria for filtering data, authors specify that xenobiotic-related features should be present in all samples. This requirement assumes of a case and control, limiting application of the workflow to human samples without defined exposures.

- We have addressed this point as part of the new studies that we have added. Specifically, we've demonstrated through an extension of the workflow how prescreening the UHPLC-MS data can be used to define 'case' (or 'exposed', as used in the manuscript) and 'control' labels for the 3 intensity-based filters, hence providing the capability to reveal unknown exposure to (suspect) xenobiotics in human samples. See new section "Untargeted workflow discovers fate of xenobiotics in humans".

2) Suggestion to consider the published literature when evaluating the confidence of identified xenobiotics (i.e. Schymanski et al. 2014 EST. 48(4): 2097.)

- We had reported confidence in annotations using the Metabolomics Standards Initiative framework. We acknowledge the recommendation, and have now furthered the reported confidence levels of identified xenobiotics by also using the Schymanski method in the manuscript and in Supplementary Tables 5, 6 and 19, as well as in the tables associated with the new section (Supplementary Tables 22-27).

3) Two case studies are presented. Each case evaluates rats exposed to a single drug with biotransformed products presented. To demonstrate how this type of exposomic workflow could be leveraged, it would be useful to apply the method to samples with unspecified exposure mixtures.

- We would first like to re-emphasise that the workflow was not initially developed as a "exposomic workflow", as we explained above, with our original manuscript making no such claims. However, recognising the importance of the reviewers' comment, we've demonstrated application of the workflow to reveal exposure to xenobiotics and their biotransformation products in human plasma samples. A whole new section ("Untargeted workflow discovers fate of xenobiotics in humans") has been developed and added to the manuscript. Specifically we present the discovery of 8 chemicals of emerging concern and 46 biotransformation products of these chemicals within the plasma of individuals.

Reviewer 3:

This fascinating manuscript describes a workflow for "untargeted toxicokinetics" using ultra-high performance liquid chromatography mass spectrometry (UHPLC-MS). As opposed to traditional toxicokinetics in which the chemical analysis focuses on a parent compound and, perhaps, a few key metabolites, this method allows identification and tracking of a xenobiotic signature of multiple metabolites. The authors discuss how this method would allow for metabolite discovery, inform evaluation of metabolism in vitro, and enhance analysis of the impact of chemicals on endogenous biochemistry.

I found the manuscript to be both scientifically valuable and generally very interesting. This is a complex analysis and I hope that most of my comments are taken as suggestions for enhancing the ease of understanding by a broader audience but not as negative comments. First, the various sections read a bit like loosely connected vignettes rather than as a single narrative. I think this is because there are many implications to the method and, to the authors credit, they have explored several.

My major comments follow:

To help with understanding everything that was done, I suggest adding two tables. First, a table describing the two primary chemicals studied might be helpful: potential columns might include # of predicted metabolites, # of analytes probable with targeted methods (that is, with available standards), # of analytes detected with workflow presented, estimated half-life in relevant species. Second, and more importantly, a table describing the datasets analyzed would be very helpful. I would appreciate known the reference, chemical, in vitro/in vivo, species, doses, measured time points, and the reason for the analysis (such as “Development of signature”, “Evaluation of time course methods”, “Evaluation of metabolic in vitro competency”) all in one place.

- We agree these are helpful summary tables. The suggested tables have been added to the supplementary information as Supplementary Table 1 and 4 and have been referred to at appropriate locations in the main text.

One more extreme approach to simplify the paper would be removing the sections related to subtracting the xenobiotic “fingerprints” to enhance analysis of effects on endogenous chemicals. Intuitively this is an important application and I am glad that the authors discuss it, but I do not know that they actually demonstrate enhanced biological understanding here. Alternatively, the authors could provide contrast with analysis of the data without the xenobiotic signature removed to see if statistical insights are improved.

- As stated by the reviewer, we agree this is an important application, hence the authors would rather not remove the section entirely. We believe it's very important to ensure that readers are provided both components of the total picture, i.e., that knowledge can be extracted from both 'toxicokinetic' and 'toxicodynamic' data within an untargeted metabolomics experiment, simultaneously.
- Instead we have investigated the reviewer's alternative suggestion, to contrast the analysis of the endogenous data with and without the xenobiotic signature to more clearly determine if the statistical insights are improved. Specifically, we have demonstrated by applying principal component analysis to the datasets – with and without xenobiotic signature – how the dataset can be misinterpreted if the xenobiotic signature is not identified and subsequently removed prior to endogenous data analysis and interpretation. This is depicted in Figures 5a and 5b and Supplementary Figures 10 and 12 and associated explanatory text.
- Further to this, we have investigated the endogenous response measured in the plasma, presented this in the new Figure 5c, Supplementary Figure 15, and Supplementary Tables 7, 8, 13, and 14, and drawn on these results, and their consistencies to the correlation analysis and univariate statistics of the cardiac tissue responses, to strengthen the evidence for the proposed biomarkers of structural cardiotoxicity.

The following comments are minor:

I think the word “extract” (as in 168-169) may be confusing to some readers, especially since one of the analyses is removing the xenobiotic signature from the data. I suggest that the authors find an alternative (such as “identify features” or “isolate for analysis”). Perhaps with slight rephrasing they might use the term “feature extraction”.

- **The authors are fully committed to clear terminology. As such, we have replaced the word “extract” with alternative suitable terms, including “discover”, “reveal” and “isolated” when referring to finding xenobiotic-related features. We have made this change 19 times.**

On lines 299-300 I think the authors should be more explicit that they are contrasting conventional quantification (that is targeted analysis of 7(?) chemicals) in contrast to methods under study here.

- **We have now clarified that the contrast is targeted analysis of parent xenobiotic only, vs. measurements of parent xenobiotic made whilst simultaneously measuring biotransformation products and thousands of endogenous features.**

On line 330 – “days 1 and 4” should be “days 1 and 14”

- **The data (displayed in Fig. 4b) show that sunitinib levels increase between days 1 and 4. We cannot infer that levels increase to day 14, as no measurement was made at that time point.**

On 342 how many biotransformation products are tracked? The 22 from Figure 2f?

- **Yes, the 21 compounds presented in Figure 2f. A citation to the figure has been added to make this clearer to the reader.**

For Line 374 How were the “putative xenobiotic-related features” identified? I presume this is the signature from the in vivo data. However, since you discussing how much of that signature was observed in vitro (section Contrasting the metabolic competencies...) having a list of the total numbers would be helpful. This could be addressed in the second table I suggested.

- **The ‘putative xenobiotic-related features’ are those isolated by the 3 intensity-based filters of our untargeted ADME/TK workflow when applied to the full peak matrix. Removal of these generates the ‘Filtered (endogenous) peak matrix’ (see Figure 1 and associated text). Statistical analyses querying the endogenous response are carried out on the ‘Filtered (endogenous) peak matrix’. Peak matrices are study, biological sample type (i.e., plasma or cardiac tissue) and assay specific, thus, e.g., only putative xenobiotic-related features discovered in HILIC positive rat plasma are removed to generate the HILIC positive rat plasma filtered (endogenous) peak matrix. As per the reviewer’s suggestion, to make it clearer to the reader how many features are removed, we now cite Supplementary Table 2 in lines 185, 267, 382, 409, 414, and 442 and have added new rows to this table detailing the number of features retained and present within the filtered (endogenous) peak matrices which are used in the statistical analyses. Note, the filtered (endogenous) peak matrices undergo further filtering and processing in preparation for statistical analyses. These steps are detailed in the Materials and Methods - section “Endogenous metabolite and lipid-based data analysis”**

REVIEWER COMMENTS

Reviewer #1 (Remarks to the Author):

Dear Editor,

Thank you for giving me the opportunity to evaluate the revised manuscript.

This reviewer appreciates the effort put into the revision and the new experiments performed on human plasma. Especially the additional information on underlying chromatographic peaks and mass spectra in the supplementary information is well received. However, the title is still misleading as also the new data exclusively focus on pharmaceuticals and not a diverse set of xenobiotics (which would have been accessible in the acquired data). Thus, I strongly suggest replacing the word 'xenobiotics' in the title with 'drugs' or 'pharmaceuticals'. Since reviewer #2 had a very similar impression of the title and content, this clear terminology is obviously important.

Remaining comments:

- It is surprising that the plasma experiments were only performed in positive ESI mode although the applied Tribrid instrument is capable of fast polarity switching and most biotransformation products ionize far better in negative ESI.

-Too little information is reported to understand the background of the samples. What kind of plasma was used, how was it stored, was information on their lifestyle and medical background available (given that so many drugs were found)? It was not stated if a pooled plasma sample was employed for QC purposes as is state-of-the-art as far as I could see.

- I do not agree with the new conclusion that '... Thus, our approach revealed insights into the exposomes of the tested individuals.' The authors looked explicitly at drugs and 'exposome' intrinsically means to investigate and report many different chemical exposure classes which was clearly not done here. Please replace 'exposomes' with 'drug exposure' which is correct.

- It is frustrating to see that only the MS2 spectra of the parent compounds will be shared with the community and not the biotransformation products which would be far more novel and interesting. I understand that it is difficult if companies are involved but I see no reason why this data should not be shared, especially when publishing in an open-access journal.

- This reviewer appreciates that the raw data was uploaded onto the MetaboLights repository but is still unable to access and judge its quality since it is not online - nearly two years after the original submission

- The rationale why the authors intentionally did not apply (at least some) reference standards for biotransformation products to enhance identification confidence (when commercially available) is not clear and I would have preferred to see some confirmation

Point-by-point response to reviewer's comments

Below, in black, we copy the comments made by reviewer 1. Please see our response to each comment in blue.

This reviewer appreciates the effort put into the revision and the new experiments performed on human plasma. Especially the additional information on underlying chromatographic peaks and mass spectra in the supplementary information is well received. However, the title is still misleading as also the new data exclusively focus on pharmaceuticals and not a diverse set of xenobiotics (which would have been accessible in the acquired data). Thus, I strongly suggest replacing the word 'xenobiotics' in the title with 'drugs' or 'pharmaceuticals'. Since reviewer #2 had a very similar impression of the title and content, this clear terminology is obviously important.

Thank you for your comments. While the authors believe the workflow presented can have much wider impact, (i.e., to discover the fate of many types of xenobiotics including, for example, industrial chemicals that have been used in deliberate exposure settings such as the safety testing of chemicals), and not just pharmaceuticals, we recognise that its demonstration was limited to pharmaceuticals within this manuscript. We have made changes to the terminology used throughout the manuscript to reflect this, including in the title, as requested.

Remaining comments:

- It is surprising that the plasma experiments were only performed in positive ESI mode although the applied Tribid instrument is capable of fast polarity switching and most biotransformation products ionize far better in negative ESI.

Selection of analytical assay, including ionisation mode, was based on the results presented in earlier sections of the manuscript (i.e., rat plasma analysis) which evidenced most biotransformation products were detected in HILIC positive. This rationale has been added to the Methods section of the manuscript (line 855-856). While the instrument used is capable of fast polarity switching, the established chromatography methods used in the author's research group are incompatible with this functionality since different mobile phase modifiers are used depending on ionisation mode, to optimise chromatography and data quality.

-Too little information is reported to understand the background of the samples. What kind of plasma was used, how was it stored, was information on their lifestyle and medical background available (given that so many drugs were found)? It was not stated if a pooled plasma sample was employed for QC purposes as is state-of-the-art as far as I could see.

We have addressed this comment and provided additional information such that the background of the samples may be better understood by the reader. This includes the kind of plasma, how it was stored and information on donor lifestyle and medical background, as suggested (lines 769-771). The use of a pooled sample as intra-study QC was conducted in our new study, as was already stated in the manuscript, see lines 823-825 of the revised manuscript.

- I do not agree with the new conclusion that '... Thus, our approach revealed insights into the exposomes of the tested individuals.' The authors looked explicitly at drugs and 'exposome' intrinsically means to investigate and report many different chemical exposure

classes which was clearly not done here. Please replace 'exposomes' with 'drug exposure' which is correct.

The authors agree the findings in this study were limited to revealing exposure to pharmaceuticals and their biotransformation products only. While this is an element of the exposome, we have edited the statement in question to make it more specific, referring now to the "pharmaceutical exposome".

- It is frustrating to see that only the MS2 spectra of the parent compounds will be shared with the community and not the biotransformation products which would be far more novel and interesting. I understand that it is difficult if companies are involved but I see no reason why this data should not be shared, especially when publishing in an open-access journal.

The authors are most certainly open to sharing data with the community and have now added spectra corresponding to biotransformation products to the MassBank repository (accessions: MSBNK-UOB-XB000xxx, available to view here <https://github.com/MassBank/MassBank-data/pull/208>). We apologise for not having done this sooner.

- This reviewer appreciates that the raw data was uploaded onto the MetaboLights repository but is still unable to access and judge its quality since it is not online - nearly two years after the original submission

The authors apologise for the delay in making the data available. The repository has since been curated and is available to view by the editor and reviewers of this manuscript using this link (<https://www.ebi.ac.uk/metabolights/reviewer018916ec-40fc-4c7e-a685-168d71c63e75>).

- The rationale why the authors intentionally did not apply (at least some) reference standards for biotransformation products to enhance identification confidence (when commercially available) is not clear and I would have preferred to see some confirmation

This was a discovery-based project thus it was not known what biotransformation products would be detected prior to completion of the data analysis. That being said, we appreciate the comment and have addressed it by adding evidence of MS/MS spectral matches to data obtained for reference standards stored in public spectral databases (mzCloud, MassBank), thus providing greater confidence (i.e., Schymanski Level 2a) in the identification of biotransformation products discovered where possible (Supplementary Figure 20).

REVIEWERS' COMMENTS

Reviewer #1 (Remarks to the Author):

No further comments, I think it is a nice piece that will be appreciated by the community.